



# Assessing the impact of future altimeter constellations in the Met Office global ocean forecasting system

Robert R. King[1], Matthew J. Martin[1], Lucile Gaultier[2], Jennifer Waters[1], Clément Ubelmann[3], and Craig Donlon[4]

[1]Met Office, Exeter, UK
[2]OceanDataLab, Plouzané, France
[3]DATLAS, France
[4]ESTEC, ESA, Noordwijk, Netherlands

**Correspondence:** Robert R. King (robert.r.king@metoffice.gov.uk)

**Abstract.**

Satellite altimeter measurements of Sea Level Anomaly (SLA) are a crucial component of current operational ocean forecasting systems. The launch of the SWOT wide-swath altimeter mission is bringing a step change in our observing capacity with 2-dimensional mesoscale structures now able to be observed over the global ocean. Proposals are now being considered

for the make-up of the future altimeter constellation. In this study we use Observing System Simulation Experiments (OSSEs) to compare the impact of additional altimeter observations from two proposed future satellite constellations. We focus on the expected impact on the Met Office operational ocean analysis and forecasting system of assimilating an observation network including either 12 nadir altimeters or 2 wide-swath altimeters.

Here we show that an altimeter constellation of 12 nadir altimeters produces greater reductions in the errors for SSH, surface

currents, temperature and salinity fields compared to a constellation of 2 wide-swath altimeters. The impact is greatest in the dynamic Western Boundary Current regions where the nadir altimeters can reduce the SSH RMS error by half, while the wide-swath altimeter only reduces this by one-quarter. A comparison of the spatial scales resolved in daily SSH fields also highlights the superiority of the nadir constellation in our forecasting system. We also highlight the detrimental impact spatially-correlated errors could have on the immediate use of wide-swath altimeter observations. However, we still achieve

promising impacts from the assimilation of wide-swath altimetry and work is ongoing to develop improved methods to account for spatially-correlated observation errors within our data assimilation scheme.



## 1 Introduction

Measurements of sea level anomaly (SLA) from satellite altimeters have been available for more than 30 years and are routinely assimilated into ocean reanalysis and forecasting systems (Le Traon et al., 2017; Davidson et al., 2019). They play a crucial role in constraining models of the mesoscale ocean in these systems, allowing estimates of the past, current and future state

of the ocean to be made. Until recently, the SLA data from altimeters were only measured along a single track directly under the path of the satellites. The nature of these nadir altimeters means that the smallest scales which can be resolved along-track are between about 35 km and 55 km depending on the altimeter (Pujol et al., 2023). While multiple satellites are available, there is still not enough data to fully constrain the mesoscale ocean due to the large gaps between tracks on any particular day. Ballarotta et al. (2019) estimated the spatial and temporal scales of maps of SLA created using three nadir altimeters, with the

mean effective spatial resolution at mid-latitudes estimated to be about 200 km.

The Surface Water and Ocean Topography (SWOT) satellite mission (Morrow et al., 2019) was launched in December 2022 with near-real time swath data products expected to be available sometime in 2024. It is the first satellite mission to measure SLA across a 120 km wide swath. There is a 20 km gap in the middle of the swath, in the centre of which a nadir instrument measures the SLA. SWOT is expected to observe the 2-D structure of mesoscale ocean processes down to between 15–30 km

in wavelength, depending on the sea-state, but will have a 21-day repeat time so will only revisit the same ocean features on relatively long time-scales. SWOT is an experimental mission to demonstrate the concept of swath altimetry and is expected to provide very useful information which can be assimilated into ocean forecasting systems to improve the accuracy of their analyses and forecasts.

There are plans to include more than one wide-swath altimeter (WiSA) on board the Sentinel-3 Next Generation (S3-NG)

operational mission which is likely to be flying in around 2030. Different combinations of nadir and swath altimeters are being considered by ESA for S3-NG and the aim of the work described here is to contribute information about the impact of different altimeter constellations on operational ocean forecasts. The main two options which are studied here include a constellation of 12 nadir altimeters, and a constellation of two WiSA satellites.

The impact of potential future observations on data assimilation systems is traditionally assessed by running Observing

System Simulation Experiments (OSSEs, Fujii et al., 2019) and that is the approach taken in this study. In an OSSE a "nature run", usually a high-resolution model run without data assimilation, is used as a representation of the true ocean. Observations are simulated using the nature run fields for both the existing observing systems as well as the new data type. This simulation of observations includes a representation of the errors expected from each observation type. The simulated observations from the existing observing systems are assimilated into a model which is usually different to the one used to generate the nature

run (different initial conditions, different surface atmospheric forcing, and sometimes a different resolution), to generate a "control" experiment. Another experiment is run assimilating the simulated observations for the existing observing systems and the new observing system. Both the control run and the OSSE run can be compared to the nature run fields in order to assess the reduction in error expected from assimilating the new data in addition to the existing observing systems. See for example Halliwell et al. (2017) for a complete description of the approach.



Recent examples of the application of OSSEs to study the impact of SWOT data include the work of King and Martin (2021) in a regional high-resolution (1.5 km) ocean forecasting system, and the work of Benkiran et al. (2021) and Tchonang et al. (2021) in a global 1/12° resolution ocean forecasting system. King and Martin (2021) included experiments showing the impact of including correlated observation errors in the simulated data since the real SWOT data is expected to contain spatially correlated errors.

One of the limitations of the OSSE approach is that the results are dependent on the model, data assimilation scheme and other aspects of the experimental set-up (such as the realism of the simulated observation errors). Also, operational ocean forecasting systems are continually developed, so the impact in today's systems will likely be different to the impact in the systems when data from future missions are actually available. To address the forecasting-system dependence of the results, coordinated OSSEs using more than one system can be carried out (see for example Martin et al. (2020)). The project to which

this work contributes developed a coordinated framework and the OSSEs were run using the Met Office ocean forecasting system (described here) and the Mercator Ocean International ocean forecasting system (described by Benkiran et al., in prep).

    The Forecasting Ocean Assimilation Model (FOAM) is the Met Office's ocean forecasting system and consists of global and regional configurations. The global ocean/sea-ice system (Aguiar et al., 2024) is run at 1/12° resolution while a regional system around the UK is run at 1.5 km resolution (Tonani et al., 2019). A lower resolution (1/4°) version of the global configuration

is used as part of the Met Office's coupled numerical weather prediction (NWP) system (Guiavarc'h et al., 2019). This is used to produce ocean reanalyses and as part of the Met Office's coupled seasonal prediction system (GloSea, MacLachlan et al., 2015). These forecasting systems assimilate SLA data from nadir altimeters, sea surface temperature (SST) data from satellites and in situ platforms, in situ profiles of temperature and salinity from various sources including Argo and gliders, and sea-ice concentration data from satellites (in the global systems only). These data are assimilated using the NEMOVAR

data assimilation software (Waters et al., 2015; Mirouze et al., 2016) together with the NEMO ocean (Madec et al., 2022) and CICE sea-ice models (Hunke and Lipscombe, 2010). In this study we consider the impact of different SLA constellations on the global configurations.

    Details of the OSSE framework used in this study are described in section 2 including the nature run, the simulation of observations, the ocean model and data assimilation system used in the assimilation experiments, and details of the experiments.

Section 3 presents the results of the OSSEs in terms of the impact of the different altimeter constellations on the sea surface height (SSH), surface currents, temperature and salinity. It also describes an assessment of the impact on the constrained time and space scales. While the results largely focus on the 1/12° global system, a comparison of the results from the 1/4° and 1/12° resolution versions of the system is also presented. Most of the results are from the assimilation of WiSA data without the expected correlated observation errors, but we show some results from experiments which include these correlated errors

in the 1/4° system to highlight some of the issues expected from more realistic data. Finally a discussion of the results and our conclusions are given in section 4.



## 2   Experiment design

Observing System Simulation Experiments (OSSEs) require several components: a nature run, observations simulated from the nature run, and additional OSSE runs using a different model setup into which the simulated observations are assimilated (Hoffman and Atlas, 2016; Halliwell et al., 2017). The nature run provides the truth against which other experiments are assessed and from which we sample observations. In this section we describe the nature run used here, the simulation of the observations, the model and data assimilation system used in the OSSEs, and the experimental set-up.

### 2.1   Nature run

To draw useful conclusions about changes to the observing system, the nature run needs to be a realistic representation of the ocean. Additionally, the OSSEs must use an ocean model which differs in enough respects so that there is sufficient error growth between the nature run and OSSEs to emulate the differences between an operational system and the real ocean. Ideally, an OSSE would use a separate lower-resolution ocean model with different parameterisations, different surface boundary inputs, and be initialised from a different (though realistic) state.

In this work the nature run was chosen to be a 1/12° global free-running NEMO model (with no data assimilation), the same resolution as the OSSEs, taken from Mercator Ocean and described by Gasparin et al. (2019) and Lellouche et al. (2018). The nature run model differs from the OSSE model in the version of NEMO (version 3.1 for the nature run), the parameter settings used, the surface forcing and the initial conditions. The nature run was initialised on 11th October 2006 from the EN4 dataset (Good et al., 2013) and was forced at the surface by atmospheric fields from the real-time atmospheric analysis produced at ECMWF. Here we consider the period starting in January 2009. The realism of the nature run in terms of large-scale variability was assessed by Gasparin et al. (2018).

### 2.2   Simulation of observations

#### 2.2.1   Standard observation types

The standard set of observations currently assimilated operationally include in situ and satellite SST data, satellite nadir-altimeter along-track SLA data, in situ profiles of temperature and salinity, and satellite sea-ice concentration (SIC) data. Simulated observations were generated for each of these data types using model data from the nature run described above. The positions of the observations were taken from real-world observation coverage representative of a recent period. Realistic observation errors were generated for each type using methods described by Mao et al. (2020) for SST and SIC, and by Gasparin et al. (2019) for in situ profiles.

#### 2.2.2   Altimeter observations

As mentioned in the introduction, the main two options for S3-NG altimeter constellations considered here are i) a constellation of 12 nadir altimeters flying in equidistant Sentinel-3A/B-like orbits and ii) a constellation of two WiSAs (each with its own



nadir altimeter). We also include a baseline altimeter constellation consisting of 3 altimeters meant to be representative of the current Sentinel-3 satellites (A and B) and Sentinel-6. The two S3-NG constellations are assimilated in addition to the nadir altimeter data from Sentinel-6. The data assimilated in the different experiments is summarised in Table 1. Other satellite
altimeters are also likely to be producing data at the same time as S3-NG and Sentinel-6, but since we do not know their likely characteristics we focus only on the Sentinel altimeters.

Simulated data of SLA were generated for the existing nadir altimeter constellation as well as for the WiSAs using the SWOT simulator described by Gaultier et al. (2016). The nature run data described above were fed into the simulator and it generated observations with realistic errors. The error budget for the baseline nadir altimeters was based on the errors in the
real datasets for Sentinel-3A, Sentinel-3B and for Jason-3 (which is representative of Sentinel-6). The nadir altimeter data were generated with a sampling of 6 km, similar to the near-real time product assimilated in our operational systems. For the constellation of 12 nadir altimeters, data were simulated assuming the same error budget as for the Sentinel-3 satellites and also with a sampling of 6 km.

For WiSA, the error budget was generated using a simulation for each component of the error expected from SWOT, in-
cluding phase error, roll error, timing error, KaRIn noise, baseline dilation error and residual path delay error from the wet troposphere correction (see Gaultier et al., 2016). Each of these components of the error are available separately in the files, so experiments could be run with different error components. In the main OSSE experiments described below, only the spatially uncorrelated components of the error budget (the KaRIn noise and the residual path delay error) were included in the assimilated data. This is a major simplification and we discuss the impact of this later in the paper. The simulated WiSA data were
generated for the 120 km wide swath (with a gap in the middle of 20 km) at 2 km resolution and were averaged to generate "super-observations" at 10 km resolution, similar to the resolution of the model grid. Since the WiSA data are expected to suffer from problems in high wave conditions, observations were removed when the significant wave height (SWH) was greater than 8 m (a wave model was also generated in conjunction with the main nature run which provided additional inputs to the SWOT simulator). For real data it is likely that the accuracy of the data will suffer at lower SWH than this threshold (see Peral et al.,
140  2015).

An example of the observational coverage of SLA from the two scenarios studied is shown in Fig. 1. This figure shows that the WiSA data increase the coverage of the ocean on each day, and that there are regions within the swath that are measured with high spatial resolution, but that significant gaps still exist. While the WiSA data measure regions of the ocean with high spatial resolution, there is a trade-off between the two scenarios in terms of spatial and temporal coverage, and particularly in
the regularity of the coverage in time and space.

## 2.3  Ocean model and data assimilation scheme

The model used in the OSSEs is NEMO at version 3.6 coupled to the CICE sea-ice model. The configuration of the ocean model is called GO6 and is described in detail by Storkey et al. (2018) while the sea-ice model configuration is called GSI8.1 and is described by Ridley et al. (2018). This ocean/sea-ice model configuration is the version currently used operationally
in the global FOAM system. It is available at two different resolutions, one at 1/4° resolution (called ORCA025) and one at





**Figure 1.** Example observation coverage from the two altimeter constellation scenarios (left: 12xnadir, right: 2xWiSA). Global 1-day coverage is shown on the top row, along with 1-day (middle) and 7-day (bottom row) coverage over the Gulf Stream region.



1/12° resolution (called ORCA12). The main results presented in this work are from runs of the ORCA12 system, but some additional experiments have been run using ORCA025 to investigate the impact of correlated errors in the WiSA observations.

The data assimilation system used in FOAM is called NEMOVAR (Waters et al., 2015) and is an incremental multivariate 3DVar-FGAT (First Guess at Appropriate Time) scheme. A key feature of the scheme is that it uses physical balance relation-

ships to transfer information between the different model variables, as described by Weaver et al. (2005). This means that SSH observations are used to estimate corrections to the model's SSH, but also affect the subsurface density field and the horizontal velocities. Similarly, observations of the subsurface temperature and salinity are used to adjust the model's density structure and affect the SSH and horizontal velocities. These different observation types are all assimilated together, and the best analysis produced to make use of them all at the same time.

Another aspect of NEMOVAR is that the information at observation locations is spread spatially (horizontally and verti-cally) to produce corrections to the model fields at surrounding locations. This is done through the so-called background error correlations, and these are modelled efficiently in NEMOVAR using an implicit diffusion operator (Weaver et al., 2016). In the operational FOAM system, the horizontal correlations are specified using a combination of two functions with distinct length-scales as described in Mirouze et al. (2016). Recent work has shown that the altimeter assimilation produces improved

results without the longer of these two length-scales, and an update to the FOAM system is being prepared which removes the longer length-scale for temperature (Carneiro et al., 2024). This change is used in the experiments described here.

Satellite observations often suffer from biases and NEMOVAR includes the facility to correct these observation biases for SST data (as described by While and Martin, 2019) and for SSH (as described by Lea et al., 2008). For SSH data the bias correction in FOAM consists of two aspects. The first is that the mean dynamic topography (MDT) needed to compare the

observations of SLA with the model's SSH contains errors which result in a bias in the observations. In the idealised OSSE set-up used here however, we produce observations of SSH directly without the need of an external MDT. The estimate of this MDT bias term was therefore initialised to zero and will account for differences in the mean SSH in the two versions of ORCA12 (for the nature run and OSSEs) used in the experiments. At the end of the experiments described later, this MDT bias was small in both magnitude and scale, as expected. The second component of the SSH bias correction is related to errors

in the pre-processing of operational SLA observations which remove some of the high frequency signals, particularly at mid-to high-latitudes (poleward of 40°). The simulated SSH data produced using the SWOT simulator removes these signals in a similar way to the real observations, so we retain the second bias correction term in these experiments.

The FOAM system uses a one day time window for the data assimilation. All observations valid on a particular day are collected and read into a one-day forecast of NEMO, during which the model counterparts of the observations are calculated

at the nearest model time-step. The observations and their model counterparts are then read into NEMOVAR, together with gridded information about the model state for use in estimating the multivariate balance relationships, and information about the background and observation error covariances. The analysis increments generated by NEMOVAR (the corrections to the model state) are then read into another run of NEMO over the same day, during which a fraction of the increments are added in on each time-step using Incremental Analysis Updates (IAU; Bloom et al., 1996).



## 2.4 Experimental set-up

The OSSE period was chosen to be from January through to July 2009. Three main runs were carried out with the FOAM-ORCA12 model and assimilation system described in the previous sub-section. The first is the Control experiment which assimilated all the standard observations, including a representative nadir altimeter constellation consisting of Sentinel-3A, 3B and Sentinel-6. The second is an experiment which assimilated data from an additional 12 nadir altimeters, in addition to the Sentinel-6 altimeter and the standard observations assimilated in the Control, which we call NADIR. The third is the 2WISA experiment which assimilated the simulated data from two WiSAs described in section 2.2.2 in addition to the Sentinel-6 altimeter and the standard observations assimilated in the Control.

The Control, NADIR and 2WISA experiments were started from the same initial condition, which was different to that used in the nature run. This came from a previous reanalysis of the FOAM system valid on 1st January 2009. A three-week spin-up of the system with assimilation of the simulated standard observations was carried out and the Control, NADIR and 2WISA experiments then started from the same initial conditions on 21st January 2009. These experiments were forced at the surface by atmospheric fields coming from the ERA-5 reanalysis produced by ECMWF (Hersbach et al., 2020) which are different to those used in the nature run (which used the real-time ECMWF atmospheric analysis). A summary of the different experiments is given in Table 1.

**Table 1.** Experimental set-up. The standard observations include satellite and in situ SST, in situ profiles of temperature and salinity, and sea-ice concentration.

| Experiments | Model configuration | Atmospheric forcing | Assimilated observations | | | | |
|---|---|---|---|---|---|---|---|
| | | | Standard obs | S6 | S3A&B | 12xS3 | 2xWiSA |
| Nature run | NEMOv3.1/LIM, ORCA12 | real-time ECMWF | - | - | - | - | - |
| Control | NEMOv3.6/CICE, ORCA12 | ERA-5 | Y | Y | Y | - | - |
| NADIR | NEMOv3.6/CICE, ORCA12 | ERA-5 | Y | Y | - | Y | - |
| 2WISA | NEMOv3.6/CICE, ORCA12 | ERA-5 | Y | Y | - | - | Y |

The Control experiment has a globally averaged SSH root-mean-square error (RMSE) of just over 6 cm (as shown in Fig. 2). This is consistent with the level of error in the operational FOAM system, estimated by comparing the one day forecasts with observations. Globally averaged RMSE for temperature in the Control peaks at about 0.8°C at 100 m depth and decays to around 0.2°C at 1500 m depth, which is also similar to the operational FOAM system. The salinity RMSE in the Control experiment peaks at the surface with a much larger value (over 1.5 psu) compared to the errors in the operational FOAM system which are estimated to be of the order of 0.25 psu at the surface (compared to Argo observations). This discrepancy is likely due to the sampling differences of comparisons with the Nature run (which measures the error everywhere), as opposed to the Argo salinity data which do not sample well the regions of highest salinity variability, e.g. in river outflows, at very high latitudes, and in regional seas. Nevertheless, the overall levels of RMSE in the Control experiment are consistent with the





errors seen in our operational system, so the idealised experimental framework used here can be viewed as representative of

the real system.

## 3 Results

In this section, we detail the impact of assimilating the simulated observations by comparing each experiment with the truth provided by the full 4-D fields from the Nature run. This grid-point-by-grid-point comparison allows us to examine the impact of the assimilation over the full domain meaning we are not hindered by incomplete observation sampling as would be the case

with a system assimilating real observations. We first describe the impact of each experiment on the SSH, the 3D temperature and salinity, and the surface currents. We then describe the impact on the temporal and spatial resolution of SSH features in each experiment and the differences seen when running the experiments with a lower resolution model. Finally, we also show the impact of the expected large spatially-correlated errors in the WiSA data.

### 3.1 Impact on SSH

The globally-averaged impact on SSH of assimilating additional altimeter observations is significantly positive in both the NADIR and 2WISA experiments (see Fig. 2). A greater reduction in the SSH RMSE is apparent in the NADIR experiment where the percentage reduction reaches ∼16% compared to ∼10% in the 2WISA experiment. The Western Boundary Current (WBC) regions dominate the SSH variability and, as shown in Fig. 3, are the regions where the additional altimeter observations have the greatest impact. Broadly, the impact is again greater in the NADIR experiment compared to 2WISA. While both show

some small regions of degradation under the Antarctic sea-ice, the 2WISA experiment additionally shows a confined region of degradation in the north-eastern Pacific in the SSH RMSE compared to the Control.

Focussing on the Gulf Stream region (Fig. 4), we see that the NADIR experiment shows a significant reduction in RMSE from the 12 nadir altimeters compared to the Control's three nadir altimeters with almost no areas of degradation. In contrast, while the 2WISA experiment reduces the SSH RMSE overall, and significantly in some places, there are also areas along the

main Gulf Stream path and its extension where the SSH RMSE is degraded. This experiment assimilated one nadir altimeter in conjunction with the two WiSAs which results in an uneven spatial sampling on any given day (as shown in Fig. 1). While the WiSA data will allow a good analysis of the SSH in the vicinity of the data on a particular day, there will be a number of days at any given location which are not sampled by the data, during which time the errors in the model will grow. This makes it much harder for the data assimilation to constrain the mesoscale eddy field at all locations with the 2WISA constellation,

compared to the constellation of 12 nadir altimeters assimilated in the NADIR experiment, which while having a less detailed picture of the SSH in particular locations, has a more even sampling on each daily assimilation cycle.

### 3.2 Impact on subsurface T/S

The assimilation of 12 nadir altimeters significantly reduces the globally averaged RMSE for temperature profiles compared to the Control with only three nadir altimeters, as shown in Fig. 5. This reduction happens at most depths, but is particularly strong



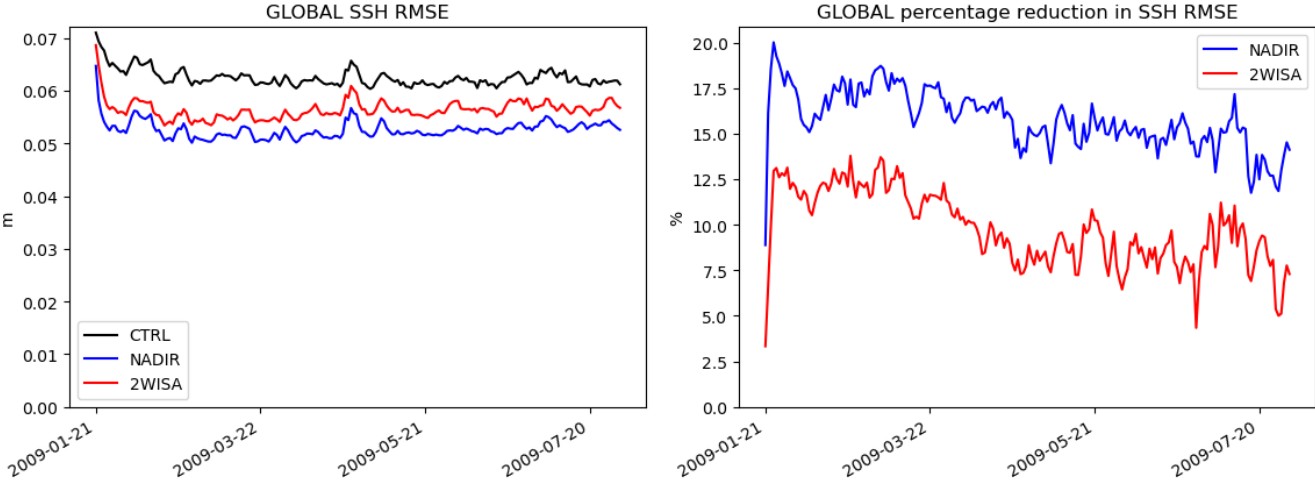

**Figure 2.** Globally-averaged SSH RMSE for the 1/12° experiments (left) along with the percentage reduction in the SSH RMSE compared to the Control experiment (right).

between about 100 m to 1000 m with a peak percentage reduction of ∼8% at 400-600 m. In contrast, the reduction in global RMSE for temperature profiles when assimilating the data from two WiSA instruments is only ∼1-2% in the upper 600 m. This is probably due to the regions where we saw degradations in the SSH RMSE (e.g., in the north-east Pacific and parts of the ACC) also having degradations in the impact on temperature profiles, which offsets any improvements seen elsewhere. In the Gulf Stream region where there is an overall improvement in SSH RMSE in the 2WISA experiment there is also an

improvement in the temperature profile RMSE at all depths above about 1000 m, peaking at ∼8% at about 600 m depth. The NADIR experiment has an even larger reduction in temperature RMSE in this region with improvements of over 20% at the same depth.

The global salinity RMSE is not significantly affected by the assimilation of either of the additional altimeter constellations. There is a slight improvement of <0.25% in the NADIR experiment relative to the Control at most depths, but a slight degra-

dation of <0.25% in the 2WISA experiment. However, in the Gulf Stream region, the impact of the additional observations is positive at depths of ∼300-900 m with up to a 25% reduction in salinity RMSE in the NADIR experiment and ∼8% in the 2WISA experiment.

### 3.3 Impact on surface currents

The assimilation of altimeter data produces corrections to the geostrophic component of the velocities and so, given the SSH

improvements we saw in section 3.1, we would expect to see improvements to the geostrophic currents but no significant impact on the ageostrophic component of the velocities. Therefore, here we follow the procedure we have used for the other variables and compare the daily mean surface velocities from the OSSEs to the Nature run.



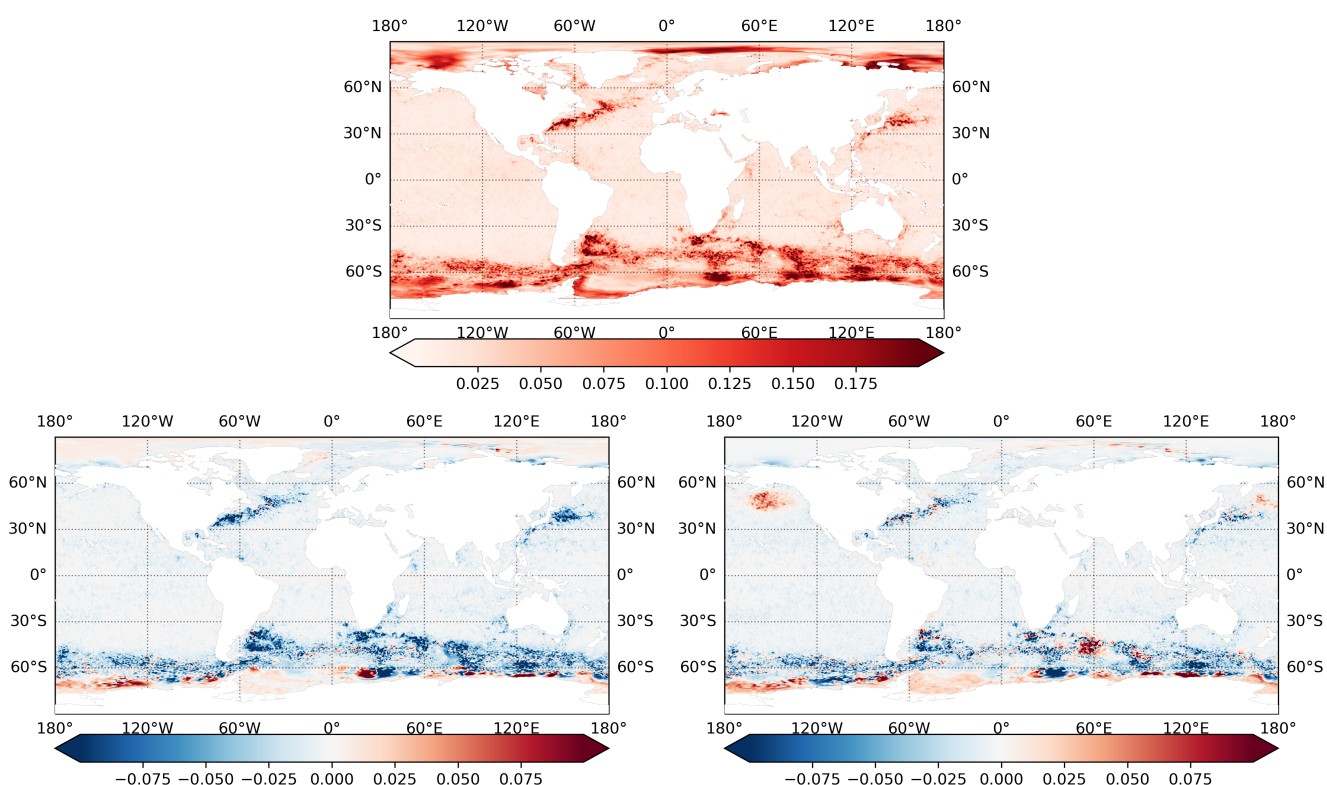

**Figure 3.** Monthly SSH RMSE from July 2009 for the Control run (top) and the difference in RMSE compared to the Control for the NADIR (bottom left) and 2WISA (bottom right) runs for the $1/12°$ experiments. Negative values imply a reduction in RMSE for the experiment (NADIR or 2WISA) compared to the Control.

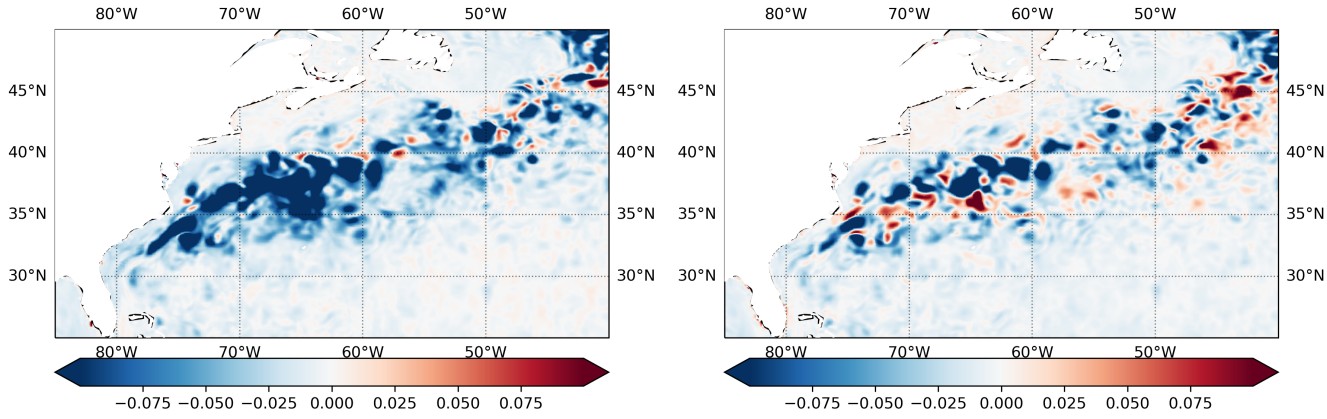

**Figure 4.** Monthly SSH RMSE difference from July 2009 compared to the Control for the NADIR (left) and 2WISA (right) runs for the Gulf Stream region. Negative values imply a reduction in RMSE for the experiment (NADIR or 2WISA) compared to the Control.





**Figure 5.** Global (top) and Gulf Stream region (bottom) profiles of the percentage improvement in RMSE compared to the Control experiment for temperature (left) and salinity (right) for the NADIR and 2WISA runs in the 1/12° system.







**Figure 6.** Globally-averaged time-series of the u- and v-components of surface current velocity RMSE for the Control, NADIR, and 2WISA runs (top), and the corresponding percentage reduction in RMSE compared to the Control (bottom) in the $1/12°$ system. Note that the y-axis in the upper plots does not start from zero to help highlight the differences between the experiments.

The time-series of globally averaged RMSE for the u- and v-components of the surface current velocity in Fig. 6 shows that the Control run velocities take about 4 months to reduce to a stable level whilst the velocity RMSE in the NADIR and 2WISA experiments reduces to a stable level within about a month. The NADIR experiment has a consistently lower RMSE than the 2WISA experiment for both the u- and v-velocity components throughout the experiments. Overall, in the final 2 months of the experiments, the NADIR experiment has 9% and 11% reductions in u- and v-velocity RMSE respectively, while the 2WISA experiment has 4% and 7% reductions in u- and v-velocity RMSE respectively.

Focussing on the last month of the experiments (July 2009) in Fig. 7 we see a similar pattern in the changes in RMSE over the globe for surface currents to the one we saw for SSH, though there are more regions of degradation in the surface currents




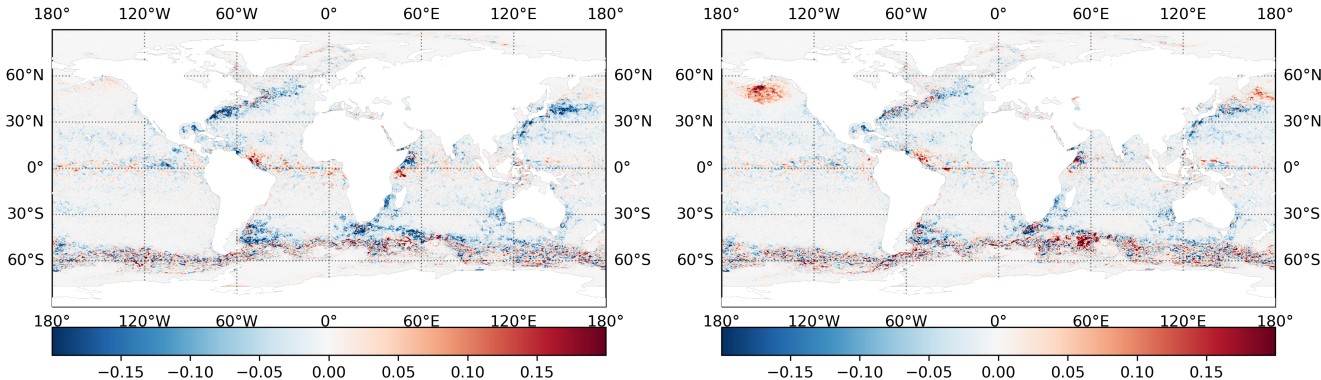

**Figure 7.** Monthly surface current speed RMSE difference from July 2009 compared to the Control for the NADIR (left) and 2WISA (right) runs.

than for SSH. In the Amazon outflow and Somali Current regions for instance there are some degradations in the surface current RMSE. The assimilation scheme used in FOAM does not generate balanced velocity increments near the equator so assimilation of extra SLA data in those regions could generate a model response which degrades the velocities. This appears to be the case in both the NADIR and 2WISA experiments, though the regions of degradation are quite small overall.

Looking in more detail at the Gulf Stream region, the position of the Gulf Stream is noticeably different between the Nature Run and the Control, particularly along the coasts of North and South Carolina (see top panel of Fig. 8). This is improved in the 2WISA run, but is more significantly improved in the NADIR experiment. The misplacement of the main Gulf Stream path in the Control and the differing improvements in the NADIR and 2WISA runs is highlighted by the dipole in the surface current mean error seen in each experiment (see second row of Fig. 8). The magnitude of the dipole reduces in the 2WISA run

(compared to that in the Control), but is smaller again in the NADIR experiment.

In addition to the change in the position of the main Gulf Stream path, there is the expected misplacement of individual eddies in the Control compared to the Nature Run. For instance, in the Nature run there are two eddies present in the southwest part of this area to the east of the main Gulf Stream path. However, only one of these eddies is resolved in the Control run. Both eddies appear in the 2WISA run, but with more diffuse structure than in the Nature run, while both eddies are resolved

in the NADIR run and are qualitatively similar to the Nature run. Similarly, in the northeast corner of this area where there is more variability than closer to the coast, it appears that the Control run cannot sufficiently initialise the position of mesoscale features. This is somewhat improved in the 2WISA run, but the NADIR run shows a much improved qualitative match to the Nature run. It also clear from the maps of the surface current RMSE and the reduction in the surface current RMSE (see bottom two rows of Fig. 8) that there is an improvement across almost this entire area in the NADIR experiment, while the 2WISA

experiment shows significant regions of slight degradation compared to the Control.




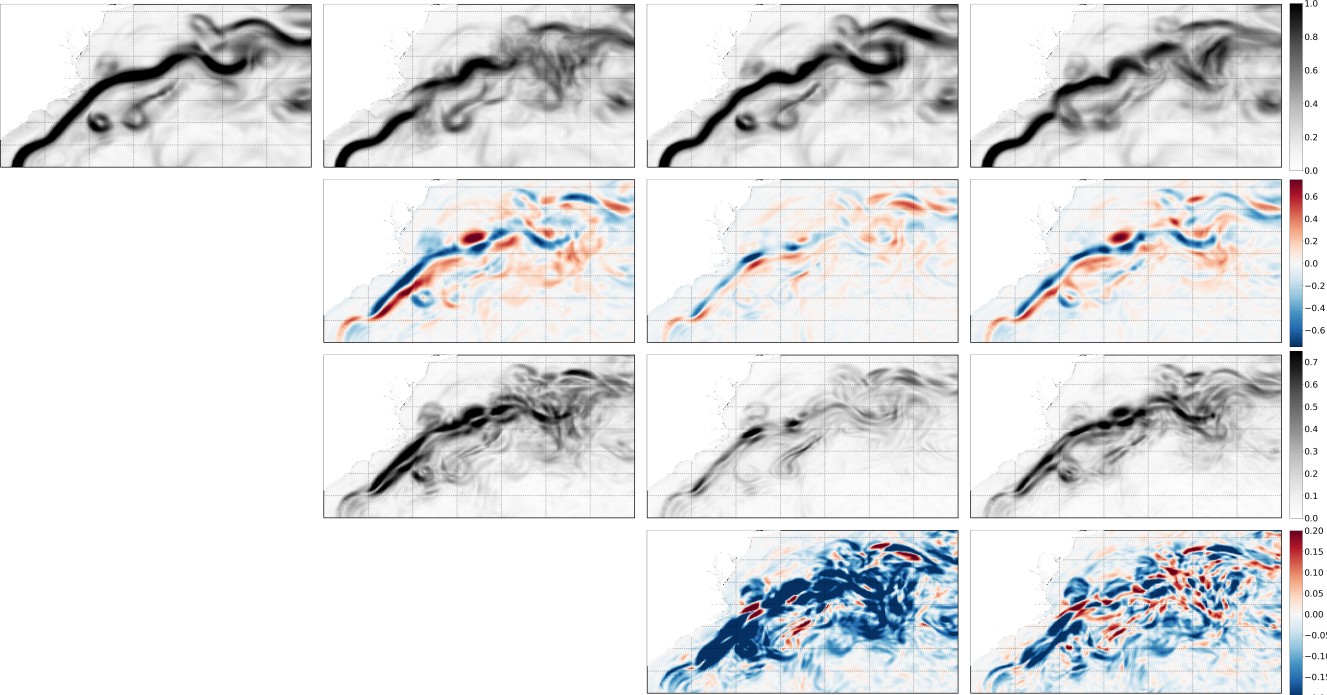

**Figure 8.** Comparison of the monthly (July 2009) mean surface current speed (top row) in the Gulf Stream region near the North American coast for the Nature Run (top left), Control (2nd column), NADIR (3rd column) and 2WISA experiments (4th column). Also shown is the surface current speed mean error (second row) and RMSE (third row) for the Control, NADIR and 2WISA experiments along with the difference in the surface current speed RMSE compared to the Control for the NADIR and 2WISA experiments.

## 3.4 Impact on constrained scales

As mentioned earlier, the WiSA satellites will measure the ocean with high resolution within their swath, but the repeat cycle is long (21 days) so that their ability to constrain the ocean dynamics on different time and space scales is not obvious a priori. To investigate how the assimilation of the different altimeter constellations constrains the model's SSH at different time and space scales the power spectra of the errors in the OSSEs were calculated, with a focus on the Gulf Stream region. This was done using the method and software of Ballarotta et al. (2019) and used the daily SSH fields from the full 6-month duration of the OSSEs.

Fig. 9 shows the power spectral density (PSD) score which compares the power spectrum of the SSH error (i.e., the difference between a specific OSSE and the Nature Run) with the power spectrum of the Nature Run SSH for the Control, NADIR and 2WISA experiments. A PSD score of 0.5 corresponds to the space-time scales where the signal is twice the magnitude of the error and is used here to define the boundary between resolved and unresolved space-time scales. All the experiments have large errors at small time-scales and short length-scales, while the large time and space scales are well constrained (have low errors). However, there is a clear improvement in the constrained scales in both NADIR and 2WISA experiments compared to





the Control. For the larger spatial wavelengths of 4° and above there is a significant improvement in the time-scales constrained
by assimilating the 12 nadir constellation compared to assimilating the two WiSA satellites. However, there is an indication that
at spatial wavelengths of 2–4° the 2WISA experiment constrains the errors at smaller time-scales than the NADIR experiment,
though the results are rather noisy in this part of the spectrum.

To further investigate the ability of the two different altimeter constellations to constrain the ocean dynamics, we used the
technique of Ballarotta et al. (2019) to calculate an 'effective spatial resolution' using the daily mean SSH fields from each
day of the experiments. The method used emulates what can be done with real observations and analysis/forecast fields of
SSH. This involves estimating the spatial resolution by calculating the ratio of the power spectral density of the SSH error
(the difference between the OSSE and the simulated observation) to the power spectral density of the observations. For the
observations, we used the simulated AltiKa observations before any errors were added, i.e., they were samples of the truth from
the Nature Run. The gridded SSH data from each experiment were interpolated to the positions of the AltiKa observations.
The along-track and interpolated OSSE data were then split into overlapping 1500 km segments every 300 km. Finally, the
globe was segmented into 10° x 10° boxes every 1° and all segments within each box were used to compute the power spectral
density. The effective resolution at each point was then given by the wavelength where the ratio described above was 0.5.

The daily effective spatial resolution is shown in Figure 10 for the Control, NADIR and 2WISA experiments. Also shown is
the gain in effective resolution in the NADIR and 2WISA experiments compared to the Control. The somewhat arbitrary term of
'effective resolution' is useful here to compare similar experiments with the same metric. However, as discussed by Ballarotta
et al. (2019), the scale of resolved ocean features is generally around one-quarter of this 'effective resolution', suggesting that
a spatial resolution of 100 km corresponds to resolving features of around 25 km diameter.

The zonal average of the effective resolution over the Atlantic basin (shown in Figure 11) highlights the difficulty in con-
straining the ocean dynamics near the equator in all experiments. The additional observations in the NADIR and 2WISA
experiments cannot change this very close to the equator. However, in both the NADIR and 2WISA experiments, there is a
clear improvement in the spatial resolution at mid- to high-latitudes with a gain in resolution of up to around 50%. As with
the earlier results, the NADIR experiment appears superior overall with a larger gain in the resolution than in the 2WISA
experiment across all ocean basins. This superior improvement in the NADIR experiment is particularly evident in all of the
Western Boundary Currents.

## 3.5 Impact of model resolution and correlated errors

To understand the impact in the different operational configurations we run at the Met Office, we repeated the experiments
described above (Control, NADIR, and 2WISA runs) using our 1/4° resolution ORCA025 system. Similarly to the ORCA12
experiments, these were initialised using a reanalysis of the ORCA025 system valid on 1st January 2009 with a 3-week spin-up
where there was assimilation of the standard set of observations. Each experiment was then started from 21st January from the
same initial condition.

The absolute SSH RMSE was higher in each of the ORCA12 experiments compared to the lower resolution (ORCA025)
counterpart, as expected due to the double penalty effect (Rossa et al., 2008). However, the percentage reduction in SSH RMSE





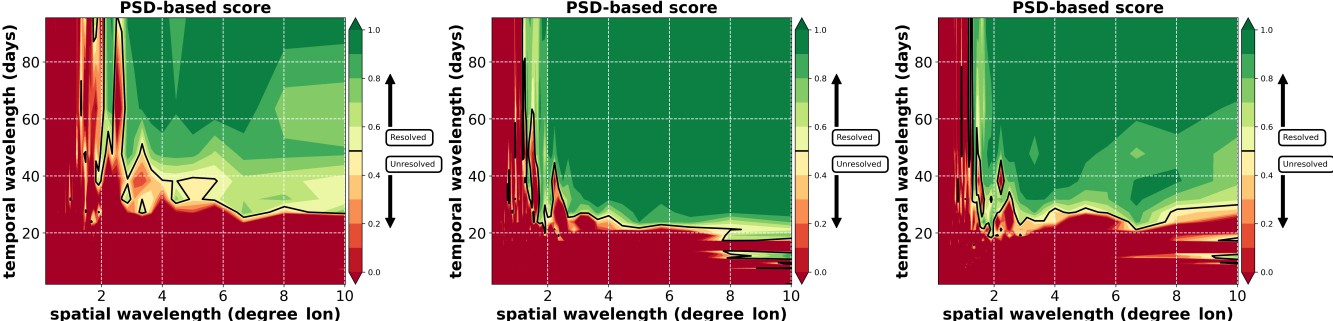

**Figure 9.** Frequency-wavelength powerspectral density score for the Gulf Stream region SSH for the Control (left), NADIR (middle), and 2WISA (right) runs in the 1/12° system. The 0.5 contour defines a boundary between the resolved and unresolved scales.

shows the NADIR experiment to be superior to the 2WISA experiment in both the high and low resolution systems. Although the absolute improvement in RMSE is largest in the low resolution system, in both systems the percentage reduction is 60%

greater in the NADIR experiment than in the 2WISA experiment. Specifically, while the NADIR and 2WISA experiments reduced the SSH RMSE compared to the Control by 16% and 10% in the experiments with the 1/12° system detailed above, the reductions were 28% and 18% respectively in the experiments with the 1/4° system (see Fig. 12).

Unlike for SSH, the velocity RMSE is of similar magnitude in the Control run of both the high and low resolution systems. Initially the ORCA025 NADIR experiment is slightly superior to the 2WISA experiment but by the final 2 months of the runs

there is a similar impact of 2/7% reduction in the u/v RMSE showing that the higher resolution system is better able to constrain the surface current RMSE with additional observations and the NADIR experiment observations allow a greater reduction than the 2WISA observations.

The impacts of assimilating wide-swath observations presented so far have used simulated observations which included only the uncorrelated KaRIn errors and residual wet troposphere correction errors, but none of the expected correlated errors.

For these additional experiments observations were again simulated using the SWOT simulator tool of Gaultier et al. (2016) and included estimates of the phase, roll, timing, and baseline dilation errors in addition to the KaRIn errors and residual wet troposphere correction errors described in Section 2.2.2 and in King and Martin (2021). Although techniques are being developed to minimise the magnitude of these correlated errors in real data, such methods may be more limited when applied to near-realtime data which is required for operational ocean prediction. Consequently, these experiments aim to highlight the

potential limitations due to correlated errors.

To investigate the potential impact of these correlated errors we have performed three experiments assimilating wide-swath altimeter observations with the 1/4° resolution ORCA025 system. 2WISA assimilates the uncorrelated wide-swath observations, 2WISA_CORR assimilates wide-swath observations which include the full correlated errors, and 2WISA_CORR_TRIM assimilates the observations with correlated errors but discards observations in the outer half of the swath where the correlated

errors are largest.





**Figure 10.** Effective spatial resolution of the daily SSH fields from each of the experiments (Control Run top, NADIR middle left, 2WISA middle right) along with the gain in the effective spatial resolution compared to the Control for the NADIR (bottom left) and 2WISA runs (bottom right).

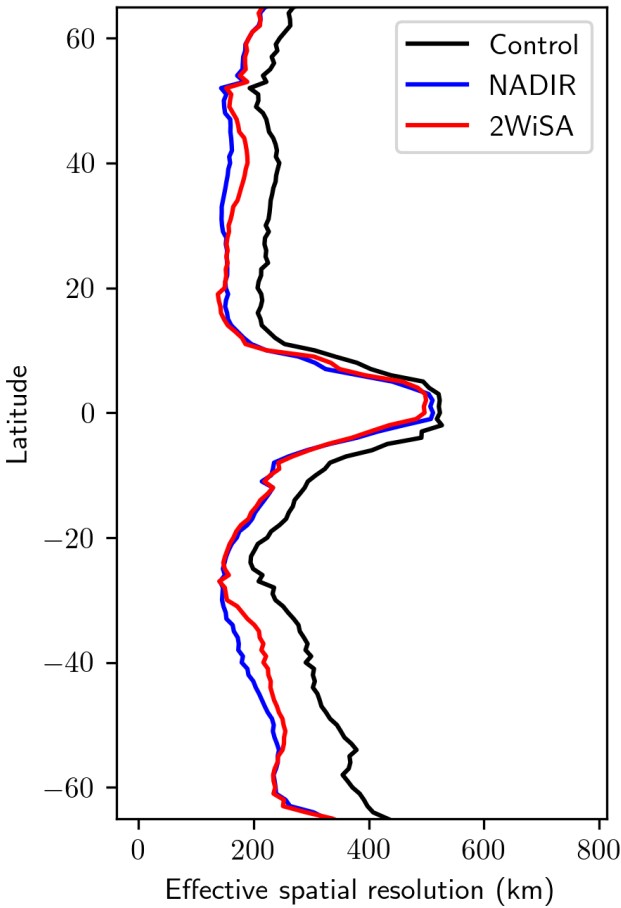

**Figure 11.** Zonal average over the Atlantic basin of the effective spatial resolution of the daily SSH fields from each of the experiments.

While the assimilation of simulated observations from 2 wide-swath altimeters without correlated errors (2WISA) reduced the SSH RMSE by 28%, when the full correlated errors were applied this reduction was only 5%. Furthermore, when only observations from the inner half of the swath were used, to discard the largest magnitude correlated errors, the reduction was still only 6%. For surface currents, the addition of the correlated errors in the wide-swath altimeter observations results in worse performance and in fact degrades the RMSE compared to the Control even when the observations are restricted to the inner half of the swath. The u/v RMSE was reduced by 2/7% when assimilating observations without correlated errors, but increased by 2/1% when assimilating the restricted observations with correlated errors, and increased by 6/8% when assimilating the full swath with correlated errors. This highlights the need to develop improved methods to represent correlated observation errors in data assimilation systems such as those of Guillet et al. (2019).



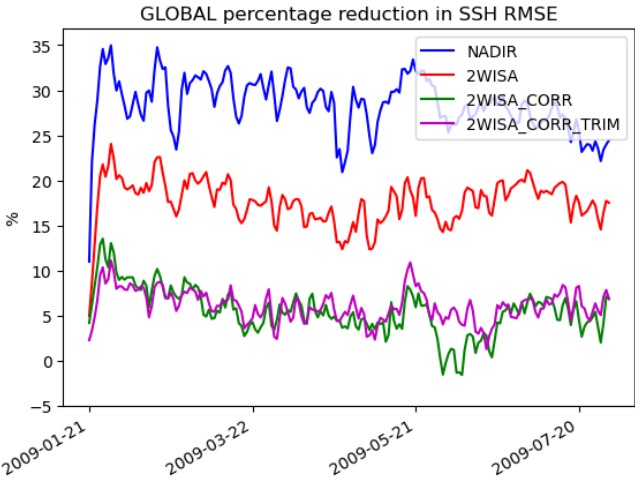

**Figure 12.** Globally-averaged percentage reduction in SSH RMSE for the 0.25° experiments compared to the Control.

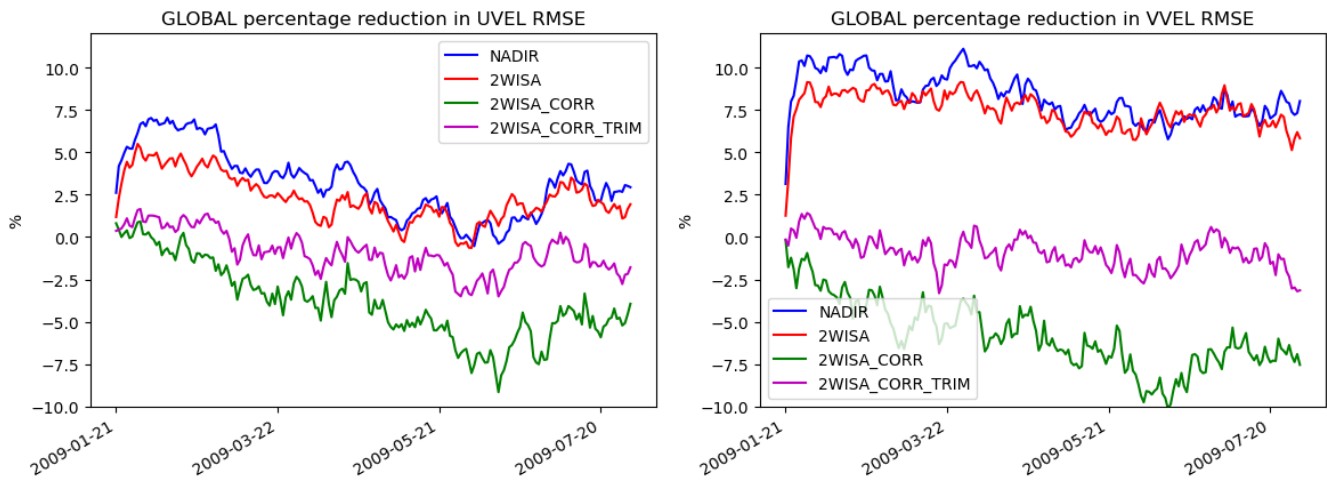

**Figure 13.** Globally-averaged time-series of the percentage reduction in RMSE compared to the Control for the u- (left) and v-components of surface current velocity for the 0.25° experiments.



## 4 Discussion and Conclusions

With the recent launch of the SWOT wide-swath altimetry mission and the on-going planning of the future altimeter constellation, we have used an Observing System Simulation Experiment (OSSE) to investigate the potential impact of two proposed next generation altimeter scenarios in an operational ocean analysis and forecasting system. While OSSEs have some important limitations, they allow us to assess and prepare for the assimilation of new and additional observations within operational systems.

In this study we found that the assimilation of additional altimeter observations has a clear positive impact in both experiments with additional nadir altimeters (NADIR) and additional wide-swath altimeters (2WISA). The SSH RMSE is reduced by 10% in our 2WISA experiment compared to the Control, while the NADIR experiment has a superior reduction in SSH RMSE of 16%. The greatest impact from the additional observations is seen in dynamic regions such as the Western Boundary Currents, for example in the Gulf Stream region the SSH RMSE is reduced by 24% in the 2WISA experiment and by 49% in the NADIR experiment. A similar impact is seen on the temperature and salinity fields where the NADIR experiment is generally superior to 2WISA. While the NADIR experiment shows a reduction in the global temperature RMSE of ∼8% at around 500m, the reduction is only 1–2% in the 2WISA experiment. The global salinity RMSE was not significantly improved or degraded relative to the Control in either the NADIR or 2WISA experiments. However, the impacts were again more pronounced in the Western Boundary Currents with reductions in the Gulf Stream region of up to 8 and 20% for temperature RMSE and up to 8 and 25% for salinity RMSE in the 2WISA and NADIR experiments respectively.

The global surface current fields again show a similar impact with the NADIR experiment having a consistently lower RMSE for both the u- and v-components of surface velocity throughout the experiments. By the final 2 months, the NADIR experiment has 9% and 11% reductions in the u- and v-velocity RMSE, while the 2WISA experiment has 4% and 7% reductions in u- and v-velocity RMSE respectively. A comparison of monthly mean surface current fields in the Control, NADIR and 2WISA experiments to those from the Nature Run showed that the NADIR experiment better corrected the Gulf Stream path and better initialised the positions and strengths of individual eddies both closer to the the North American coast and in the Gulf Stream extension region further from the coast. The improvement in the initialisation of mesoscale features is an important aspect expected from the increased observation coverage from either altimeter constellation. Knowledge of the surface currents is important for a number of users of operational ocean forecasts, but there is a significant gap in our ability to observe and constrain this field. Until there are satellite observations of the total surface current vectors, altimetry will remain our most important tool for constraining surface currents in our forecasting systems.

A comparison of the spatial and temporal scales resolved in each of the experiments highlighted a clear improvement in both the NADIR and 2WISA experiments compared to the Control. The temporal resolution of features appears superior in the NADIR experiment for length-scales greater than 4° while at smaller spatial scales the 2WISA experiment appears to constrain the errors at shorter time-scales than the NADIR experiment. Focussing on the spatial resolution of daily SSH fields showed a clear improvement over the Control at mid- to high-latitudes with a gain in resolution of up to around 50%, with the NADIR





experiment again superior overall with a larger gain in the resolution compared to 2WISA, which was particularly evident in the Western Boundary Currents regions.

While the 2WISA experiment showed on overall lower impact than the NADIR experiment, a specific region of degradation was found in the northern Pacific affecting the SSH and surface current RMSE fields. We have found that the component of our SLA bias correction scheme which is intended to correct for errors in the surface forcing (particularly at high latitudes) inhibited the assimilation of the wide-swath altimeter observations in this region. Although we were unable to find why this affected only the 2WISA experiment and not the NADIR, we found that the effect was localised to the region of degradation

in the northern Pacific and did not affect the impact of assimilation in other regions, nor the overall order of the impact or our conclusions. The additional observations assimilated in the two experiments differ in a number of ways which could contribute to the different impacts we have found: the gridding of the observations differs (as explained in Sect. 2.2.2), although both are at slightly higher resolution than the model grid-scale; the wide-swath altimeter provides the ability to resolve across-track features; and the spatial and temporal gaps between the tracks/swaths differs markedly.

Although the experiments described here show that a superior impact is found when assimilating the 12-nadir altimeter constellation, similar experiments by the Mercator Ocean International group find the opposite result (Benkiran et al., 2024), that is in their system a greater impact is found when assimilating the 2-wide-swath altimeter constellation. An important difference between our system and the Mercator system is the time window used in the data assimilation schemes. The Met Office system uses a 1-day assimilation window, compared to 7-days in the Mercator system. Therefore, the most important

difference between the two sets of additional observations may be the regularity of the sampling. While the total number of observations in a single data assimilation window is similar between the two altimeter constellations, the NADIR experiment with 12 nadir altimeters sacrifices high spatial resolution for higher temporal resolution and more even sampling, while the 2WISA experiment has the high spatial resolution, but with much larger gaps between individual swaths.

The main experiments reported here were performed without spatially-correlated errors included in the simulated wide-swath

altimeter observations. However, such correlated errors are expected in real wide-swath data and so additional experiments were run including these error components. The calibration of observations before public release may be able to address major components of these error correlations, but it is not yet clear how effective this will be, nor whether it will be possible for the near real-time data products required for assimilation into operational systems. Our experiments showed that while the assimilation of simulated observations from 2 wide-swath altimeters without correlated errors (2WISA) reduced the SSH

RMSE by more then 25%, when the full correlated errors were applied this reduction was only 5%. We further explored the impact of assimilating only the inner portion of the swath where correlated errors are lower, but the effect was minimal with an SSH RMSE reduction of 6% compared to the Control. The impact of the inclusion of correlated errors was greater for surface currents where we found the u/v RMSE was reduced when assimilating observations without correlated errors by 2/7% for the u- and v-velocity components, but was increased by 6/8% when correlated errors were included. In this case, restricting

the observations to the inner swath limited the increase in RMSE to 2/1% for the u- and v-velocities. We therefore stress the importance of developing improved methods to represent correlated observation errors in data assimilation systems (e.g., Guillet et al., 2019) and testing these with real wide-swath altimeter data. Such data assimilation developments will be a crucial



aspect in extracting the full potential of SWOT and other upcoming wide-swath altimeter missions. Additionally, we suggest that the impact of assimilating wide-swath altimetry on the surface currents should be closely monitored as this is an important
physical parameter for many users which is difficult to verify due to the sparseness of available observations (Aijaz et al., 2023).

The European Space Agency (ESA) is planning its future altimeter constellation through the Sentinel-3 Next-Generation Topography mission and have chosen to develop a solution involving both nadir and wide-swath altimeters. While the experiments detailed here show a superior impact from an additional 12 nadir altimeters compared to 2 wide-swath altimeters, we
still show promising impacts from the new wide-swath altimeters. It must also be noted that the results from OSSEs provide an estimate of the impact in the specific system tested. These results do not reflect the best achievable impact, but rather an estimate of the impact in our current system. Future model and assimilation changes tailored to the observation network is expected to lead to a greater impact. Work is now underway to prepare our operational systems to assimilate real data from the SWOT wide-swath altimeter mission, including accounting for correlated errors in the assimilation scheme, which will test our
ability to make good use of WiSA data to deliver improved ocean forecasts to users.

*Data availability.* The nature of the 4-D data generated in running the model experiments requires a large tape storage facility. These data are of the order of 300 TB (terabytes). The model data can be made available upon request from the authors.

*Author contributions.* All authors contributed to the design of the project. RRK, MJM and JW worked on the experimental design, RRK ran the experiments, LG simulated the observations, RRK, MJM, JW and CU contributed to the analysis, and the manuscript writing was led by
RRK and MJM. All authors have read and agreed to the published version of the manuscript.

*Competing interests.* The authors declare that they have no conflict of interest.

*Acknowledgements.* Funding support from the European Space Agency is gratefully acknowledged. We also acknowledge our collaborators in developing NEMOVAR at CERFACS, INRIA and ECMWF.



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
