# Peer review of "Assessing the impact of future altimeter constellations in the Met Office global ocean forecasting system"

_EGUsphere, 2024_

## Author Comment (AC1)

**Response to reviewer #1**

We thank the reviewer for their efforts in reviewing our manuscript. These queries have helped us to revise and improve our article.

Reviewer comments are shown below in black with the author response in red.

**General Comments:**

There are several issues I feel that the authors should address prior to publication:

1. A critical aspect of the study is how the errors are estimated and applied to synthetic observations and how they are specified in the assimilation system. In particular, the errors applied to observations used for NADIR and 2WISA are poorly described and lack details regarding the amplitudes of perturbations applied. The degree to which the system is constrained to the nature run will depend not only on observational coverage, but also on the observational errors applied. This could have an impact on both the RMS errors as well as the spectral properties and ability to make use of the smaller scales present in the wide swath measurements. There are several specific suggestions provided below on how this aspect could be improved.

In response to this comment and the detailed comments below (more detailed responses in-line below), we have updated the text to give a fuller explanation of how the simulated observations were produced. We have clarified what we mean by "realistic errors" by detailing the error components for the standard observations as well as the nadir and wide-swath altimeter observations. We have also included the RMS of the altimeter errors and the number of altimeter observations assimilated in each experiment to inform the degree to which each experiment is being constrained by the observations. Finally, we have updated Section 2.4 with a comparison of innovation statistics from our OSSE control with those from our operational system assimilating real observations to demonstrate that our OSSE framework provides a realistic emulation of the real system.

2. A significant degradation is found for 2WISA experiment in the Northeast Pacific Ocean as compared to the Control (with 2 nadir altimeters). The impact of this degradation is visible in many of the figures, and the authors have clearly done their best to avoid this feature in their interpretation of results (e.g. Fig. 11 that only shows results for the Atlantic Ocean). In the conclusions, the authors claim that this is related to the bias correction method for sla observations and that the impact is isolated in the northeast Pacific, but admit they were unable to explain why this occurs. I can understand that correcting and rerunning the experiments would be a costly and time-consuming effort, however, its difficult to be sure that this issue is not the cause of the reduced benefits found for 2WISA. Especially when the results presented are opposite to similar studies published previously (as noted in the conclusions). This issue is compounded by the fact that very little information is provided to describe the procedure. The authors should provide a clearer justification and description, together with some evidence to support their claim that the impact does not affect results in other regions.

We have investigated this issue further and have found that this region in the north-east Pacific has very low SSH variability in the Nature Run. This is not captured in the background errors in our data assimilation system resulting in the DA adding noise in this region. Although there are a few other areas with a similarly low SSH variability (north of the ACC and in the mid-Atlantic), the north-east Pacific feature aligns with the boundary where we stop applying the full SSH balance (where the temperature stratification is less than 5K). An inspection of daily SSH increments (and the RMS of monthly increments) from our experiments shows large length-scale increments between altimeter swaths which align with this boundary in the 2WISA experiment only.

We had previously suggested that our SLA bias correction might have caused this degradation, but on reflection the larger bias correction in this region was a symptom rather than the cause. It appears that the interaction of very low SSH variability with the transition from applying balanced SSH increments to only barotropic increments leads to spurious SSH changes in this region.

We have attempted to assess the global impact and also the impact away from this isolated feature as the aim of the work was to assess the potential impact of these observing networks in a system emulating our operational model and DA system. The impacts we have seen, including the localised detrimental impact in the 2WISA experiment, are realistic impacts expected if we assimilated real data in this way. This has allowed us to identify issues which need to be addressed when assimilating real data to make best use of the available observations. Interestingly, we have seen a similar feature in our first early experiments assimilating real SWOT data giving confidence to the realism of this degradation. However, we do not intend (and have clarified the text accordingly) to suggest that the 12 nadir altimeter constellation is inherently superior to the 2 wide-swath constellation. We have emphasised that while OSSEs provide a source of information on the potential impact of a set of observations, they are necessarily strictly applicable to that system, though can provide useful guidance more generally. This was one of the drivers for coordinating the design of the experiments with the Mercator group (Benkiran et al 2024) to explore whether the impacts were sensitive to the system used.

We have responded to the related detailed comments below and have updated Sections 3.1, 3.4 and our discussion/conclusions to explain and justify our claim that the degradation in the north-east Pacific is localised.

3. Finally, I agree with the authors conclusion that the spatial and temporal sampling differences between the 12 nadir and 2 wide swath approaches is likely the primary source of the differences presented. However, it would be helpful to illustrate this aspect more clearly. Fig. 1 presents differences in coverage between 1 day and 7 day windows. However, it would be useful to show how the 21-day repeat coverage of 2WISA affects the assimilation statistics. For example, differences could be shown for a small region (e.g. the size of spatial correlation scales applied?) highlighting the intermittence in 2WISA experiment as compared to NADIR. Are the SSH errors for 2WISA smaller than NADIR following the overpasses and then grow with time? If so, this would demonstrate that the wide-swath data is being correctly assimilated and that the reduced impact is indeed the sampling. On the other hand, if the problem is the observational error specification (or the SLA bias correction procedure), we would see that even following an overpass of a wide-swath altimeter the 2WISA experiment would fail to constrain smaller scales. Additionally, maps showing differences in

increments over the Gulf Stream region could also reveal if differences between NADIR and 2WISA are due to the presence of SLA biases requiring constant increments at each cycle to maintain the system close to the nature run as opposed to correcting chaotic turbulence (which then grows between cycles).

To illustrate the effect of the different sampling of the nadir and wide-swath altimeter observations, we have included a figure showing maps of the SSH increments from each experiment on a single day and also the RMS of the SSH increments over the 21-day repeat cycle of the wide-swath altimeters. The relatively wide spacing of the altimeter swaths in the 2WISA experiment over our 1-day assimilation window produces short length-scale increments near the observation locations and longer length-scale barotropic SSH increments in the regions between altimeter swaths. In contrast, the relatively close spacing of the altimeter tracks from the 13 nadir altimeters (in the NADIR experiment) over our 1-day assimilation window produces predominantly small-scale SSH increments. The long-term effect of this is apparent in the RMS of the SSH increments over 21-days (the repeat cycle of the wide-swath altimeter observations) where larger RMS values indicate the assimilation scheme is introducing more variability in the 2WISA experiment than in the NADIR. We have also updated Section 2.2.2 describing the errors added to each simulated observation type and note that the RMS errors included in 10km resolution ("super-obbed") wide-swath altimeter observations are ~0.5cm compared to 1.4cm in the nadir altimeter observations.

It appears that a longer assimilation window might allow the data assimilation to make more consistent changes in the NADIR and 2WISA experiments by better initialising mesoscale structures which are relatively static over the window. Although this could improve results for SSH, this would have to be balanced with the impact on other variables as in an earlier study we found than a shorter assimilation window improved results for SST with little impact on other variables (Lea et al. 2015). Given the difference in the daily SSH increments between the NADIR and 2WISA experiments we might be able to improve the impact in the 2WISA experiment by altering the balance between the short and long length-scale increments while retaining our 24-hour assimilation window.

We have addressed the detailed comments below (referring to Line235 and Line414) and have updated Section 3.1 and Section 4.

**Specific Comments:**

L20: Technically the satellites measure SSH. SLA is obtained by removing a mean SSH surface. As SSH is referred to later in the paper, it would be better to use consistent language throughout.

We have updated the text to consistently refer to SSH.

L25: The following sentence uses the term "nadir". Include it here to make it clear what this means.

Added to clarify meaning of nadir.

L39: It would be better to define WiSA when SWOT is first mentioned (or not at all). It is not clear to me the benefit of using an acronym for Wide-swath altimetry and not for nadir altimetry when the two alternative observing system approaches being considered are nearly the same length. Moreover, the use of WiSA is used inconsistently throughout the paper, with "wide-swath altimetry" used at times and WiSA other times. It is also somewhat confusing to have an acronym for WiSA and the name of one the experiments 2WISA. I would propose to remove the WiSA acronym and just say "wide-swath altimetry" and "nadir altimetry" to be clear.

Thank you for the suggestion which clarifies when we are referring to the type of observation rather than one of the experiments. We have replaced "WiSA" with "wide-swath altimeter" but retained the experiment name 2WISA.

L101: It would be good to note that the sea ice model is also different.

Now noted in text that the nature run used the LIM2 ice model while the OSSEs used the CICE model.

L112: A brief description of how the observation errors are generated should be added along with details concerning the amplitude of the errors and whether spatially-correlated errors are introduced. It would be helpful to note here the different types of error (instrumental, representativeness) and what is being estimated here.

We have added the following text to describe how observation errors are generated for the non-altimeter observations used in our experiments.

Briefly, this involved adding representation errors by randomly selecting the date either three days before or after the observation date, then using these time-shifted nature run values in the interpolation process (instead of the correct date). This produces larger errors in regions with higher variability, which is desirable for generating realistic representation errors and is the same method used by Gasparin et al. (2019) for the AtlantOS inter-comparison. Uncorrelated instrumental errors were also added to each observation. These were created by randomly sampling from a Gaussian distribution with zero mean and an appropriate standard deviation for each observation type. The standard deviations used for the synthetic SST (0.1--0.5~K) and in situ profiles (0.01--0.05 K, 0.01--0.05~psu) varied with platform type and are detailed fully in Mao et al. (2020) and Table 4 of Gasparin et al. (2019), respectively. For SIC observations only representation error was added, but this accounts for uncertainties in the marginal ice zone, so the errors are much larger in this area than elsewhere.

L124: It would be helpful to elaborate on what you mean by "realistic errors". For the baseline nadir altimeters what amplitude is applied for the error?

We have rephrased and expanded the text following this sentence to clarify what we mean by "realistic errors" by including the following text in Section 2.2.2.

"The error budget for the nadir altimeters used the error spectrum computed for current altimeter missions (defined in Esteban-Fernandez 2014) which includes components resulting from instrumental errors, the residual path delay error from the wet-troposphere correction, and the sea-state bias. The error spectrum was computed from level-3 products which already had the typical corrections applied to real altimeter observations (removal of tides, dynamic

atmosphere correction, and long wavelength error) allowing us to directly simulate level3-like altimeter data. The resulting RMS error in our simulated nadir altimeter observations is ~1.4cm."

L134: Final L3 altimetry products typically used for assimilation are corrected for many of the raw satellite errors (tides, DAC, longwave, wet tropospheric). How do you deal with this?

The error spectrum used was computed from L3 products which had already been corrected for each of these effects allowing us to directly simulate L3-like altimeter data. The text (given in response to the previous comment) included this clarification.

L134: You mention here that only uncorrelated errors are Karin noise and residual path delay error. However, later on line 344, it is noted that Karin and wet tropospheric errors are used. Also, it would be helpful to provide the amplitude of the perturbations applied to the synthetic observations to simulate errors.

For the main experiments, we chose not to include the full correlated errors in the wide-swath altimetry observations as these are expected to be removed in part by the post-processing to create real level-3 observations. However, we included additional experiments with our lower resolution system to explore the potential impacts if the correlated observations could not be accounted for. We have corrected the text (in Section 2.2.2 and later in Section 3.5) to clarify that the residual path delay is the error due to the wet troposphere correction and have added the RMS of the error components.

L135: Superobbing to 10km. But nadir is at 6km? Why not make it the same? This choice reduces the along-track resolution of the wide-swath data and could affect the extent to which small scales are constrained.

We considered matching the resolution, but these choices reflect the pragmatic choice we expect to make with real data, i.e., we will continue to assimilate nadir altimetry at the provided resolution (~6km), but plan to super-ob the dense wide-swath altimeter observations to 10km initially. The simulated nadir observations were produced at 6km along-track resolution, close to what is provided in the real L3 observations assimilated into our operational system. For these observations we decided it was best to leave this unchanged. However, the model grid spacing is ~9km, and so the nadir resolution is over-sampled. To avoid issues with overfitting with the denser wide-swath data, we chose a sampling that better matched the grid spacing. We have added some explanation in the text to note this disparity and also note that our small background correlation length-scale is limited to 25km. Due to the size of the grid-scale and the background correlation length-scale we would not expect to constrain scales smaller than this. We have updated Section 2.2.2 to detail the overall number of altimeter observations assimilated in each experiments (@188k in the Control, 831K in NADIR, and 970k in 2WISA).

L138: Is nadir data affected by SWH?

Nadir altimeter observations are much less affected by SWH than wide-swath altimetry observations due to the differing footprints and retrieval algorithms. Although the performance of SWOT in high SWH appears to be better than anticipated, the return signal is also better that what we will have on WISA type instrument and studies are ongoing to better understand and simulate the impact of large SWH on SSH retrieval.

L147: Why mention NEMO version number and not CICE version number? Please add version number for the latter.

This has now been added.

L156: Here you mention SSH observations. It would be good to be consistent with the introduction and stick to either SLA or SSH (unless there is a reason to differentiate).

We have updated the text to consistently refer to SSH.

L158: "…all assimilated together, and …". Remove comma. The rest of the sentence isn't very clear. It would be good to reword.

For clarity this has been reworded to "These different observation types are assimilated simultaneously to produce a single analysis for each 1-day assimilation window."

L164: It would be helpful to state the length scales used as they have a direct impact on how the altimetric information will affect the model solution.

The text has been updated to include these length-scales.

L175: What high-frequency errors are being referred to here? DAC? A bias should have a time mean, but it is mentioned that high-frequency signals are removed from observations by the simulator. Which is it? Additionally, problems with this bias correction term are used to explain the poor performance of 2WISA experiment in the Northeast Pacific Ocean. As such, it would be appropriate to provide additional detail. Indeed, it would be good to add a comment here regarding the role of this correction in affecting the results later on.

The second altimeter bias term referred to here is designed to account for differences between the modelled and observed SSH due to errors in the representation and removal of high-frequency atmospheric effects, particularly at mid- to high-latitudes. As the simulated SSH data were produced from a nature run which used a different source of surface forcing compared to the OSSEs, we retained the second bias correction term in these experiments. Section 2.3 has been updated to clarify.

L186: Why not run for a full year? Evaluating over only Jan. to July will create a hemispheric bias with respect to the sea ice cover (see comments regarding Fig. 3).

Unfortunately, these 1/12 degree model runs were prohibitively computationally expensive to run. Whilst a year or longer would have been preferred, we ran experiments which were long enough that our conclusions were not affected by a spin-up period. Although we did not sample every season, all OSSE experiments covered the same period and so allowed us to make a fair comparison of the impact of the two observation scenarios.

L190: Why does the 2WISA experiment not include Sentinel 3a/b? It would allow us to see the impact of adding the 2 wide swath altimeters directly. As it stands, a comparison of 2WISA and Control would include the impact of the 2 wide-swath altimeters, but also the impact of removing 2 nadir altimeters (Sentinel 3a/b). It seems the removal of these two altimeters has an important impact on the results. As such, an additional simulation with only

1 altimeter (Sentinel6) would provide a means to separate the effects (or rather, with 2 wide-swath altimeters and sentinel3a/b).

The altimeters included in each scenario were chosen to inform the decision by ESA on the future altimeter constellation. It was felt that including S3A and S3B in the control was a good representation of the existing altimeter constellation, but ESA wished to compare the two scenarios as described. Further experiments may have helped disentangle the causes of the differences found, but we were unable to run any additional experiments due to the computational expense.

L205: It is not clear to me why the authors are comparing min/max values using innovations from the operational system to min/max values using the full grid in the OSSE. Why not assess min/max values from the OSSE innovations? This way the sampling would be the same.

Thanks for the suggestion, this should have occurred to us. We have updated the text in Section 2.4 with a comparison of the innovation statistics in a system assimilating real observations against our OSSE control. Although we did not have this system running operationally over period chosen for our experiments (and so the precise observational coverage differs), this is a fairer comparison than what we had initially.

L209: It is standard practice to produce an OSE prior to the OSSE to verify the OSSE framework provides an equivalent response (e.g. when withholding altimetry data). Has an OSE been performed? For example, it would be useful to see the impact of withholding sentinel 3a/b in an OSE and in the OSSE (see comment regarding line 190). This may help to explain some of the areas of degradation seen in the 2WISA experiment.

We agree that an ideal set-up would have included OSEs in addition to the OSSEs to confirm that the systems have the same impact when an observation type is removed. However, we don't think that this is in fact standard practice in the field due to the prohibitive cost involved (this would require an OSE control, OSE experiment with some observations removed, plus the equivalent OSSE with some observations removed, almost doubling the computational cost). Instead, the comparison of the innovation statistics in our control run and operational system showed that our baseline experiment performs in a similar manner to our operational system assimilating real observations.

L220: It would be helpful to know how the total number of observations differs between NADIR and 2WISA experiments. Fig. 1 gives a qualitative sense to this, but total number of observations would give an idea how well the system is able to benefit from the information. Also, the choice to apply a "superobbing" of the data to a 10km grid will affect this number.

Thank you for this suggestion. We have updated the text in Section2.2.2 to specify that the NADIR experiment assimilated 831k altimeter observations per day while the 2WISA experiment assimilated 970k altimeter observations (after "super-obbing").

L220 (Fig. 2): It would be helpful to provide statistics for other regions, especially the Gulf Stream region since this is the focus of Fig. 4. The global statistics will be strongly affected by the strong signal found under ice (see comment below) thereby biasing the overall (ice-free) results. The global results are also affected the problem in the northeast Pacific.

We have updated Section 3.1 to include the percentage reduction in RMSE for the Gulf Stream region during the discussion of the impact in that region.

L223: (Fig. 3): Why only show monthly mean for July? Given that the statistics are quite stationary (apart from initial few analyses), using fields for the full simulation would provide more robust statistics.

We chose to focus on a single month at the end of our experiments to avoid any spin-up in the impacts. This was particularly clear in the time-series for surface currents. While using multiple months would improve the statistics somewhat, we felt this struck a reasonable balance.

L225: Why is there an impact under the sea ice? Were synthetic ssh data used even where there is sea ice? Since this is not typically done in the operational systems it should be rejected in the OSSE as well. In July there should be considerable sea ice cover in the southern ocean. As a result, a change in the SSH data sets assimilated shouldn't have an impact under the ice. If the study has used data under the sea ice this needs to be mentioned and the impact of this choice discussed in detail as it appears to have a first order impact on the study results.

Thank you for highlighting this. Synthetic SSH data were not assimilated anywhere where the model sea-ice fraction was greater than 5% to emulate the operational situation. However, SSH increments due to balanced changes from temperature and salinity were spread under the ice from observations near the ice edge. While this emulates what happens in our operational, the detrimental impact on SSH under the sea-ice had not been noted in our operational system (due to a lack of observation). These experiments have highlighted that we should restrict the spreading of this information under the ice. Section 3.1 has been updated with the following text.

"Even though we have no SSH observations in sea-ice covered areas, the long background error correlation length-scale produces changes to the (highly variable) SSH under the sea-ice. While this emulates what happens in our operational system, these experiments have highlighted that we should restrict the spreading of this information under the ice."

L226: The degradation in the northeast Pacific is quite unusual. The cause for this feature should be mentioned here as it is not associated with the wide-swath data themselves but rather a suggested problem in the SLA bias correction. Without this explanation, it suggests there is a problem in the experimental setup and undermines the reliability of the findings.

As detailed in our response to General comment #2 earlier, we have investigated this issue further and have found that this region in the north-east Pacific has very low SSH variability in the Nature Run. This is not captured in the background errors in our data assimilation system resulting in adding noise in this region. Although there are a few other areas with a similarly low SSH variability (north of the ACC and in the mid-Atlantic), the north-east Pacific feature aligns with the boundary where we stop applying the full SSH balance (where the temperature stratification is less than 5K). An inspection of daily SSH increments (and the RMS of monthly increments) from our experiments shows large length-scale increments between altimeter swaths which align with this boundary in the 2WISA experiment only.

We had previously suggested that our SLA bias correction might have caused this degradation, but on reflection the larger bias in this region was a symptom rather than the cause. It appears that the interaction of very low SSH variability with the transition from applying balanced SSH increments to only barotropic increments leads to spurious SSH changes in this region. Interestingly, we have seen a similar feature in our first early experiments assimilating real SWOT data giving confidence to the realism of this degradation.

We have updated Section 3.1 to better explain the cause of this feature and justify our claim that the degradation is localised to the north-east Pacific.

L235: It would be helpful to have a timeseries of RMS error over a small region to illustrate the point being made here about the sporadic sampling in time. We should see smaller errors following assimilation which grow between overpasses. It would also help to demonstrate the WISA data are being assimilated correctly and that it is indeed the time sampling the issue.

We looked at including a time-series of the RMS error over a small region, but found the time-series was too noisy to clearly show the effects of the sampling. However, to illustrate the effect of the different sampling of the nadir and wide-swath altimeter observations, we have included a figure showing the SSH increments on a single day and the RMS of the SSH increments over a 21-day period. We have updated the text in Section 3.1 to include the following

"The daily SSH increments shown in Fig.6 illustrate the effect of the different spatial sampling in the NADIR and 2WISA experiments. The relatively wide spacing of the altimeter swaths in the 2WISA experiment over our 1-day assimilation window produces short length-scale increments near the observation locations and longer length-scale unbalanced SSH increments in the regions between altimeter swaths. In contrast, the relatively close spacing of the altimeter tracks from the 13 nadir altimeters (in the NADIR experiment) over our 1-day assimilation window produces predominantly small-scale SSH increments. The cumulative effect of this is shown by the RMS of the SSH increments over a 21-day period (the repeat cycle of the wide-swath altimeters) in Fig.6 which indicate that the data assimilation is introducing more variability in the 2WISA experiment than in the NADIR experiment. While the wide-swath altimeter data will constrain the SSH in the vicinity of the data on a particular day, there will be a number of days at any given location which are not sampled by the data, during which time the only constraint we have on the SSH comes from the correlations with distant locations. This makes it much harder for the data assimilation to constrain the mesoscale eddy field at all locations with the 2WISA constellation, compared to the constellation of 13 nadir altimeters assimilated in the NADIR experiment, which while having a less detailed picture of the SSH in particular locations, has a more even sampling on each daily assimilation cycle."

L238: "The assimilation of 12 nadir altimeters…". In fact, the NADIR experiment assimilates 13 nadir altimeters doesn't it? (i.e. 12 + Sentinel6). It is mentioned the Control has three (2+Sentinel6), so a consistent nomenclature should be used.

This has now been corrected.

L244 (Fig. 5): Why do both experiments show a strong (20%) degradation in temperature below 1000m in the Gulf Stream region?

We have added the following explanation to Section 3.2. The balances in our data assimilation scheme allow altimeter observations of the SSH to introduce subsurface changes to the temperature and salinity. However, previous experiments have shown that the assimilation of in situ profiles and altimeter observations can sometimes work against one another (King et al. 2018). With such a large increase in the altimeter observations, the balanced changes applied to subsurface temperature and salinity may dominate over the changes due to the in situ observations leading to degradations in some regions over some depths, such as those seen for temperature below 1000m in the Gulf Stream region.

L258: Surface currents will be strongly affected by the winds applied. Would it not be more interesting to assess just below the surface (e.g. 15m depth) where a large impact on geostrophic currents should be visible? Use of 15m depth would also allow greater potential transferability of results to actual comparisons with drogued drifters.

We chose to assess surface currents due to their use in many of the operational products we supply to users. The surface currents will be strongly affected by the winds used (the source of which differs between the Nature Run and the OSSEs as detailed in Section 2.1), but this emulates the operational situation where there are errors in our knowledge of the atmospheric conditions. While using 15m currents may allow some comparison with future assessments of the impact of assimilating new altimeter observations, this would in any case be hindered by the sparse sampling of drifters relative to the full-field comparisons here.

L276 (Fig. 8): A significant difference between the experiments is in the representation of a large feature in the top right of the panels (these panels should have lat/lon labels). If we consider the panels a 7x7 grid, the feature would be in X=2:3, Y=5. The NADIR run captures well this large feature whereas the other runs look more diffuse (possibly due to higher variability?). This feature appears to be part of the Gulf Stream mean flow. This suggests that its representation may be part of an improvement in bias, rather than having to do with finer resolution.

Fig 8 was added to demonstrate that there is both a difference in the Gulf Stream mean flow and variability between the nature run and Control experiments and further that both are improved in the NADIR and 2WISA experiments, but that the improvement in the mean flow and variability appears greatest in the NADIR experiments (as discussed in Section 3.3). Lat/lon labels have now been added to this figure.

L283. Should read "It **is** also clear …"

Corrected

L291: Over what region are the PSD scores calculated? Could you define 'Gulf Stream region' more precisely as this is used loosely throughout the paper. Does this region include land (e.g. the 'Gulf Stream region' shown in Fig. 4 or the one used in Fig. 8). If not and a smaller domain with only ocean points is chosen, how are PSD scores near 10deg obtained as they would have very few cycles.

We used the area 40–80W, 30.0–50N to define the Gulf Stream region for the PSD scores. The text has been updated to include this information.

L303: Ballarotta et al. (2019) aim to assess the effective spatial resolution of their procedure to re-grid altimetry observations. The use of this technique here has a somewhat different connotation. The scale at which the PSD of the error is larger than ½ of PSD of the observations would be more appropriately referred to as the limit of constrained scales. It is correct to note that Ballarotta et al. refer to this as "effective resolution", but it should not be interpreted as so here. For example, there are areas for which the data assimilation system is having trouble assimilating the altimetry observations resulting in larger errors (e.g. in the northeast Pacific Ocean and in the Arctic around 180deg). In the latter, Fig. 10 suggests an "effective resolution" of over 500km! It would be more accurate to refer to this as the limit of constrained scales. In this case, it would highlight that the assimilation system isn't constraining the SSH to any measurably degree in this region of the Arctic.

We agree that the term "effective resolution" could be misleading and so we originally included a discussion on how this relates to what we termed the 'feature resolution'. We have moved this discussion to earlier in Section 3.4 to clarify what is being assessed here rather than renaming the metric.

L312: '…at each point'. If there are a series of 10x10deg boxes every 1deg, then the corresponding PSDs and ratios are considered to correspond to the center of the 10x10deg box. Is this what 'at each point' refers to? The center of the 10x10deg box? Perhaps reword to make this more clear.

Yes, we were referring to the centre of each box and have updated the text to clarify.

L312: '…where the ratio described above was 0.5'. Reword to something like this '…where the ratio described above is equal to 0.5', or 'crossed the threshold of 0.5'.

Corrected to "where the ratio described above was equal to 0.5".

L313: The anomaly in the northeast Pacific is glaringly obvious in Fig. 10, yet there is no comment. This unexplained issue puts in question the validity of the rest of the study.

We have updated the text here in Section 3.4 to comment on the feature and refer back to the explanation now added to Section 3.1. In fact, this metric highlights well that the north-east Pacific degradation is localised and while there is this region with a detrimental impact, we still see significant improvements generally in the 2WISA experiment, but overall the NADIR experiment shows larger improvements in our system. Again, this is not the best that can be achieved using these two sets of observations, but rather is a realistic estimate of their impact in our current system.

L351: For these three experiments is the only change the introduction of different perturbations to the synthetic data? Is any change made to the data assimilation system to account for correlated errors? Is the observation error variance changed in the assimilation system? Also, it would be helpful to provide details regarding the amplitude of the correlated errors and how they were determined. Moreover, while its noted that efforts are underway to develop corrections to these correlated errors its not clear the extent to which some correction is used here. Without any correction, the errors would be extremely high (e.g. greater than 50

cm). While the correlated errors may not be completely removed, a reasonable approximation would be to use assume some correction of the correlated errors. So what is used here and what is this estimate based on?

Yes, for these experiments the only change was to the errors included in the wide-swath altimetry observations. No change to the data assimilation systems were made for these experiments other than to attempt using only observations from the inner half of the swath in the 2WISA_CORR_TRIM experiment to discard the observations with the largest magnitude errors. We have added detail to Section 3.5 to clarify this and to describe the components and magnitude of the correlated errors added to the simulated observations. We agree that without any correction, the errors can be very large, but we aimed to assess how our system would cope in this extreme scenario.

L400 "…showed **an** overall…". Typo.

Corrected.

L404: "*Although we were unable to find why this affected only the 2WISA experiment and not the NADIR, we found that the effect was localised to the region of degration in the northern Pacific and did not affect the impact of assimilation in other regions, nor the overall order of the impact or our conclusions*". I find this issue somewhat troubling. If the authors were unable to identify the cause of the problem, how can they be sure it is localized in the north Pacific? As noted above, leaving it to the conclusions to offer an explanation for this issue undermines the reliability of the results. It would be better to note earlier in the text that this issue is present in the results and efforts have been made to provide an assessment of the impact of the satellite constellations that are not affected by this issue.

We have updated Section 3.1 and 3.4 to discuss the cause of this issue earlier in the manuscript (as discussed above in reply to earlier comments). We have also reorganised our conclusion section to separate into a discussion and conclusions. We have directly addressed the cause of the superior impact in the NADIR experiment and also the cause of the degradation in the Northeast Pacific. In our conclusions we have also emphasised that this is not the best that can be achieved using these two sets of observations, but rather is a realistic estimate of their impact in our current system.

L414: The main difference is not necessarily the assimilation window, but rather the observations used in the analysis. The system used in Benkiran et al. (2024) produces daily analyses using observations from days in the past and the future, similar to a Kalman smoother approach. Could observations from days before and following the analysis be included in the FOAM system in a similar manner? It would be helpful to clarify the text on this point.

In both systems all of the observations are assimilated. With our 1-day assimilation window each analysis uses only the observations from a 24-hour period. It does appear that a longer assimilation window may be more conducive to initialising mesoscale structures which are relatively static over the window. Whilst this could improve results for SSH, this would have to be balanced with the impact on other variables as in an earlier study we found than a shorter assimilation window improved results for SST with little impact on other variables

(Lea et al. 2015). However, given the difference in the daily SSH increments between the NADIR and 2WISA experiments we could also explore the impact of altering the balance between the short and long length-scale increments. We have updated Section 4 to include this discussion.

L420: The issue of correlated errors is an important one, but various aspects of how it is approached here require further clarification. The precise values of errors applied are not described nor is how the assimilation system is modified in consequence.

In response to the earlier comment on line 351, we have added to Section 3.5 to detail the correlated errors added and their magnitude and also to clarify that our data assimilation system was not modified. These experiments were an extreme test and a very simple approach to restrict assimilation to the inner half of the swath where the correlated errors are smaller. These experiments still show a benefit from assimilating the wide-swath altimeter observations.

L448: Craig Donlon appears as a co-author, but his contribution is not indicated.

We updated this section to clarify Craig's involvement in the design of the project.

---

## Author Comment (AC2)

**Response to reviewer #2**

We thank the reviewer for their efforts in reviewing our manuscript. These queries have helped us to revise and improve our article.

Reviewer comments are shown below in black with the author response in red.

=== General comment

This paper conducts OSSEs with a 3D-VAR-based eddy-resolving system to compare the impacts of 12 Nadir and 2 WiSA satellites on accuracy. While OSSEs are useful for evaluating various yet-to-be-constructed observation networks, this study is limited to only three experiments: assimilating standard observations, standard observations plus 12 Nadir satellites, and standard observations plus 2 WiSA satellites. To comprehensively determine the most efficient observation networks, more diverse experiments are necessary. Additionally, it would be beneficial to include information on observation coverage and funding considerations for constructing these networks.

Our aim here was to investigate the potential impact of two specific proposed observing networks to inform the planning of ESA as they explore options for the Sentinel-3 Next-Generation Topography mission. While additional experiments may have offered further insight, the computational cost of running these high-resolution systems limits the length and number of experiments. We did however include a comparison of the impacts in our high- and low-resolution systems and further experiments in the low-resolution system to further explore the impact of correlated errors.

Moreover, OSSEs produce results which are specific to the system and observation network used and can be difficult to generalise. This is made apparent by the differing results from the coordinated experiments run by the Mercator Ocean International group (reported by Benkiran et al. 2024) which we discuss in our manuscript.

As the reviewer notes in a later comment, funding considerations are another important aspect affecting the eventual choice of which observing network to implement. Given the system-specific responses, the synergy between different observation types, and external factors affecting observing platform design and deployment, we do not think that it is practically possible to optimise an observing network using OSSEs. Instead, we aimed to determine how effectively our system would be able to assimilate observations from two specific proposed networks. This allowed us to identify issues that will affect the assimilation of real wide-swath altimeter observations.

Moreover, the manuscript uses colloquial expressions, lacks fundamental details about the data assimilation systems, and does not employ statistical tests. These elements are essential for a scientific paper. Therefore, I conclude that the current paper does not meet the criteria for proceeding to the review process and expect significant revision in the next manuscript.

It is not clear to us which expressions the reviewer regards as colloquial, but we have addressed the detailed comments below on some of the phrasing used. We believe the use of the active voice rather than passive voice is a valid choice here (and in line with journal guidelines) which makes the manuscript easier to read.

We have given an overview of the data assimilation scheme used in our system in Section 2.3 with a focus on the balances and correlation length-scales used. Detail has been added on the simulation of the observation errors and how this varies between the wide-swath and nadir altimetry, and we have also included discussion on how the resulting increments differ between the experiments to understand the differing impacts. We also refer to earlier papers describing the implementation of NEMOVAR in our global forecasting system.

To demonstrate the impact of assimilating the different observation networks, we have used the standard practice for comparing OSSEs including full field differences, effect on the bias and RMSE, along with less common metrics such as the power spectral comparisons to understand the differences between the systems.

=== Specific comment

Even in the abstract, there are grammatical errors (e.g., "now able to" in L4 and "greatest" in L10) and unclear abbreviations (e.g., SWOT, SSH, RMS). Throughout the manuscript, the descriptions are written in colloquial expressions (e.g., "we see"). Therefore, it is necessary to revise the entire manuscript to ensure scientific and objective descriptions.

We do not see an issue with the phrase "now able to". For context, we wrote that "The launch of the SWOT … mission is bringing a step change…with 2D mesoscale structures now able to be observed over the global ocean.". We also do not see an issue with the word "greatest" in the sentence "The impact was greatest in…". While we could have used "largest" here that might imply a wider spatial impact, rather than a difference in the magnitude of the difference.

However, we have checked the manuscript to ensure abbreviations are defined on first use and updated accordingly.

Finally, we have used phrases such as "In Figure X, we see that…" rather than using the passive voice alternatives of "In Fig X it can be seen". We believe this is a style choice to engage a reader.

The authors use the expression "data assimilation (DA) constraints model" in this manuscript. However, DA does not make any corrections to the model source code except when it is used for parameter estimation; therefore, this expression is inappropriate.

We cannot find this specific phrase in our manuscript, though we have used phrases such as "SSH observations…play a crucial role in constraining models of the mesoscale ocean" to mean that DA constrains the model dynamics. We certainly did not mean to suggest that the DA in some way alters the model source code. We have updated the manuscript to clarify our meaning.

L32: SWOT data in 2023 became available in early 2024.

We have updated the introduction to include this point and clarify that we were referring to near-realtime observations which will be required if we are to assimilate them into realtime analysis and forecasting systems (in our case this requires the processed observations to be available at most 48 hours after the observation time).

L37: The use of "very" and similar expressions should be avoided as they lack objectivity.

Here we are describing that wide-swath altimetry observations will be "very useful". As this is a qualitative statement, we feel the use of "very" to emphasise how useful these data are expected to be is a natural and appropriate use of language. The experiments described in the paper are an attempt to quantify how useful wide-swath altimetry can be.

This study focuses only on the two SSH observation networks (two WiSA and 12 Nadir satellites) planned by the ESA. However, the observation coverages and funding required to construct these networks are substantially different. Even if the ESA plans are currently limited to these two networks, additional sensitivity experiments are necessary to determine the most efficient observation network. Since OSSEs enable the evaluation of various unconstructed observation networks, it is essential to leverage this advantage.

The funding required to build and operate these satellites is commercially sensitive and not known to the authors. However, this was a project initiated by ESA specifically to address these two proposed scenarios which are being considered by the mission advisory groups. As mentioned above, while additional experiments may have offered further insight, the computational cost of running these high-resolution systems limits the length and number of experiments. We did however include a comparison of the impacts in our high- and low-resolution systems and further experiments in the low-resolution system to further explore the impact of correlated errors.

L46: Toy models and low-resolution models such as Lorenz-96 are used in the nature run.

It is unclear to us what the reviewer is referring to here. In OSSEs, a nature run is generally the highest resolution ocean model available and is used as a representation of the true ocean. In our experiments, the nature run is a 1/12 degree global free-running model as described in Section 2.1.

The "control run" in this manuscript is included in the OSSEs. It would be better to incorporate the control run into the OSSEs and avoid using the term "control run" throughout the manuscript.

We think our phrasing on line 52 where we referred to a control run and an OSSE run might have caused this confusion. Instead, we now refer to an OSSE framework of a control run along with additional experiments. We have retained the term "Control run" as this is a standard term to refer to the baseline experiment before the addition of more observations.

L74: Better to add 3D-VAR based before NEMOVAR.

We have updated the text to clarify that we have used a 3D-Var version of NEMOVAR.

Please specify the major differences between the NEMO models used in the nature run and the OSSEs in the 2nd paragraph of subsection 2.1.

We have updated this section to list some of the major differences between the NEMO versions.

Please add a citation for "the real-time atmospheric analysis produced at ECMWF" in L104-105.

As we can find no publication describing the specific ECMWF IFS product used, we have added a footnote with the URL linking to the ECMWF real-time data.

To confirm whether the data assimilation systems are functioning correctly, it is essential to show the prescribed observation error variance and covariance. In this manuscript, however, there are only citations of previous papers and almost no specific information. This also applies to background observation errors.

Our aim here was to estimate the impact of assimilating two specific extensions to the observing network in our operational system. To do this we constructed an OSSE to reflect that operational system and made no changes to the data assimilation scheme. In Section2.3, we give an overview of the data assimilation scheme with a focus on the balances and correlation length-scales used. Detail has been added on the simulation of the observation errors and how this varies between the wide-swath and nadir altimetry, and we have also included discussion on how the resulting increments differ between the experiments to understand the differing impacts. We also refer to earlier papers describing the implementation of NEMOVAR in our global forecasting system to avoid repeating a full system description which has already been published. We have also expanded section2.4 to include a comparison of the innovation statistics from the OSSE control and our operational system (rather than comparing operational innovation statistics at observation locations with full-field statistics from the OSSE, as we had previously). We included this comparison to demonstrate that the data assimilation system was functioning similarly in the OSSE Control and in our operational system.

Please specify "since we do not … Sentinel altimeters" in L120-121.

We have updated the text as follows to clarify our meaning. "Other satellite altimeters are also likely to be producing data at the same time as S3-NG and Sentinel-6, but since we do not know their likely characteristics we focus on Sentinel-6 in conjunction with either 2 wide-swath or 12 additional nadir altimeters."

Since observation coverage significantly impacts the analysis accuracy, it is essential to indicate the differences in observation coverage (percentage) among the OSSEs.

Thank you for this suggestion. We have updated Section 2.2.2 to detail the number of altimeter observations assimilated in each experiment to augment the figures detailing the spatial and temporal sampling of the different observing networks. Our Control experiment

assimilated on average 188k altimeter observations per day. With the super-obbing applied to the wide-swath altimeter observations, our 2WISA experiment assimilated on average 970k altimeter observations per day (including Sentinel-6, the two wide-swath altimeters and the nadir altimeter component of each wide-swath altimeter). On the other hand, the NADIR experiment with Sentinel-6 and an additional 12 nadir altimeters assimilated on average 831k altimeter observations per day.

Please modify the description in L178-180 for readers to understand.

We have rephrased this as follows to clarify.

"The FOAM system uses a 1-day assimilation window, meaning that an analysis is produced daily using observations over a 24-hour period. The observation operator in NEMO is used to calculate a model counterpart to every observation at the nearest model timestep and interpolated to the observation location. The innovations (the difference between the observation and the model counterparts) are used by NEMOVAR together with gridded information about the model state for use in estimating the multivariate balance relationships, and information about the background and observation error covariances. The analysis increments generated by NEMOVAR (the corrections to the model state) are then read into another run of NEMO over the same day, during which a fraction of the increments are added in on each time-step using Incremental Analysis Updates (IAU; Bloom et al., 1996)."

In the third paragraph of subsection 2.4, it is unreasonable to compare the accuracy between the practical operational systems of FOAM and virtual OSSEs because these frameworks are completely different. It is unnecessary to compare these results, and it would be better to remove them.

We have improved this section by making a comparison of the innovation statistics from the OSSE control and our operational system (rather than comparing operational innovation statistics at observations locations with full-field statistics from the OSSE, as we had previously). We included this comparison because the aim of the OSSE was to emulate our real system as we are quantifying the impact in the simulated system to provide an estimate of the impact on the real system.

Please specify "incomplete" observation sampling in L214.

We have rephrased this to clarify our point that OSSEs have the advantage of knowing the true state at all model grid points and times unlike in reality where our knowledge of the true ocean state is limited.

"significant" and "significantly" can be used only if the statistical tests are conducted.

To avoid confusion we have rephrased where we previously used the term significant when referring to clear differences without specific statistical tests.

In this paper, the objective is to evaluate the impacts of 12 Nadir and 2 WiSA satellites on accuracy. However, most figures, especially those depicting spatial patterns, do not illustrate their differences, which is inconsistent with the stated objective.

We have chosen a variety of ways to illustrate and quantify the difference between our three experiments (Control, NADIR and 2WISA). This includes line plots of different metrics for the three cases and often then a direct comparison of the improvement with respect to the baseline scenario (Control). For figures depicting spatial differences, we have opted to show maps of the baseline metric (for example, the SSH RMSE from the Control) and then the change in that metric in the NADIR and 2WISA experiments. While further plots could have been included to show the difference between those differences, we did not feel this was necessary.

Please provide an explanation for why both the 12 Nadir and 2 WiSA experiments result in degraded SSH accuracy around the Antarctic region.

Thank you for highlighting this. Synthetic SSH data was not assimilated anywhere where the model sea-ice fraction was greater than 5% to emulate the operational situation. However, SSH increments due to balanced changes from temperature and salinity were spread under the ice from observations near the ice edge. While this emulates what happens in our operational, the detrimental impact on SSH under the sea-ice had not been noted in our operational system (due to a lack of observation). These experiments have highlighted that we should restrict the spreading of this information under the ice. Section 3.1 has been updated with the following text.

"Even though we have no SSH observations in sea-ice covered areas, the long background error correlation length-scale produces changes to the (highly variable) SSH under the sea-ice. While this emulates what happens in our operational system, these experiments have highlighted that we should restrict the spreading of this information under the ice."

Please modify the descriptions in Lin 231-233.

We have updated this section (3.1) to better illustrate the effect of the different sampling of the nadir and wide-swath altimeter observations. We have included a figure showing maps of the SSH increments from each experiment on a single day and also the RMS of the SSH increments over the 21-day repeat cycle of the wide-swath altimeters. The relatively wide spacing of the altimeter swaths in the 2WISA experiment over our 1-day assimilation window produces short length-scale increments near the observation locations and longer length-scale barotropic SSH increments in the regions between altimeter swaths. In contrast, the relatively close spacing of the altimeter tracks from the 13 nadir altimeters (in the NADIR experiment) over our 1-day assimilation window produces predominantly small-scale SSH increments. The long-term effect of this is apparent in the RMS of the SSH increments over 21-days (the repeat cycle of the wide-swath altimeter observations) where larger RMS values indicate the assimilation scheme is introducing more variability in the 2WISA experiment than in the NADIR.

Adding SSH contours to Figure 4 would enhance clarity by illustrating the positions of fronts and eddies.

We chose not to do this because this plot shows the difference between the RMSE of two experiments over a month. While fronts and eddies clearly affect the structure seen here, there may be differences between the experiments from between tracks which are not coincident with these structures and any changes will also be smoothed over time.

Please specify the reasons for the degradation of temperature and salinity accuracies in the 12 Nadir and 2 WiSA experiments.

We have added the following explanation to Section 3.2. The balances in our data assimilation scheme allow altimeter observations of the SSH to introduce subsurface changes to the temperature and salinity. However, previous experiments have shown that the assimilation of in situ profiles and altimeter observations can sometimes work against one another (King et al. 2018). With such a large increase in the altimeter observations, the balanced changes applied to subsurface temperature and salinity may dominate over the changes due to the in situ observations leading to degradations in some regions over some depths, such as those seen for temperature below 1000m in the Gulf Stream region.

In Figure 5, it would be beneficial to include the temperature and salinity RMSEs in addition to the improvement ratio. To enhance clarity, consider specifying the use of different scales on the x-axis for each panel or using consistent scales across all panels.

Given the large difference in scale for the 4 plots, we have chosen to add a note in the caption to draw attention to the different scales used on the x-axes.

Since geostrophic velocities dominate most of the global ocean, it may not be necessary to present detailed validation results of surface currents in subsection 3.3. It would suffice to only describe that the results of sea surface currents are qualitatively similar to those of SSH.

We have included an analysis and discussion of the impact on surface currents as this is an important parameter of interest to many users. Quantifying the impact is therefore useful in our view. Also, we have shown that the impact, while broadly similar to SSH, is different.

No label for the color scale in Fig. 7.

Thank you for spotting this. We have added a label to the colourbar.

Please specify reasons for the differences in spatial patterns between SSH and surface currents (Figs. 3 and 7, respectively).

Our assimilation scheme uses linearised balance relationships to account for correlations between ocean variables. However the velocity balance is not applied close to the equator resulting in some of the differences seen in the spatial patterns of SSH and surface current impacts. This is now discussed in Section 3.3.

L267: Please specify reasons why the degradation signals are not distributed uniformly across the entire equatorial regions.

The differences are largest in the Amazon outflow and Somali current regions where the climatological background errors used may not properly account for the variability in these regions for the chosen period of the experiments.

In Figure 8, it is unnecessary to display both the monthly mean errors and RMSEs.

This was included to aid the interpretation of the change in both the mean flow and variability in this region discussed in Section 3.3.

The definition of the power spectral density (PSD) score appears not to be reasonable. It's unclear whether the PSD is calculated in the spatial or temporal directions, and the rationale behind calculating the ratio between the PSD of SSH error among OSSEs and that of true SSH is unclear. Since this definition is relevant to all descriptions in subsection 3.4, I will read the remaining descriptions at the next round.

Section 3.4 describes the use of two power spectra-based metrics which use the ratio of the spectral content of the error (the SSH difference between each experiment and the nature run) and the spectral content of the true signal (from the nature run) to determine a signal-to-noise ratio. The first (shown in Fig. 10) uses the 2D frequency-wavenumber power spectra to define a PSD score which distinguishes the resolved and unresolved scales in time and space. We have updated the description in Section 3.4 including a link to the open-source code used. The second metric (shown in Fig. 11) uses the wavenumber power spectrum to determine the limit of the scales constrained in each experiment. Section 3.4 has been updated to better define the PSD-based scores.

The reasons for using models with different horizontal resolutions of 0.25° and 1/12° should be specified. If the results from both resolutions are qualitatively the same, it may not be necessary to show the results from the 0.25° resolution.

We have restructured the text in Section 3.5 to clarify that the reason for using systems with different horizontal resolutions was to explore the impact on the two operational systems we run at the Met Office. After demonstrating the impacts are similar in the two systems, we used the lower resolution (and so computationally cheaper) system to run additional experiments exploring the impact of including correlated errors in the simulated wide-swath altimeter observations.

There are no universally accepted rules for using "/" to denote interchangeable expressions, as seen in "2/7% reduction in the u/v RMSE" in L340.

We have removed this and described the reductions explicitly.

The observation error variance is likely different among the three experiments (2WISA, 2WISA_CORR_TRIM, 2WISA_CORR), indicating a failure in this study to explore the impacts of observation covariance errors.

In all of the experiments we have described, the observation and background variances were not altered from those used in our operational system and Sections 2.3 and 3.5 have been updated to make this clear. While there are many ways in which our operational system could be adapted to make best use of new wide-swath altimeter observations, we felt that this work was beyond the scope of this current project. Our aim here was to investigate the impact of two specific observing network scenarios being assimilated into a system as similar as possible to our operational system (which is the method we are employing in our first attempt to assimilate real wide-swath data from SWOT).

Generally, the discussion and conclusion should be delineated separately. Moreover, a conclusion spanning over 2 pages is excessively long. Given that it does not succinctly summarize the results, I will review this section in the next revision.

We agree that this section was too long and have reorganised to separate into a discussion and conclusions. We have directly addressed the cause of the superior impact in the NADIR experiment and also the cause of the degradation in the north-east Pacific. In our conclusions we have also emphasised that this is not the best that can be achieved using these two sets of observations, but rather is a realistic estimate of their impact in our current system.

---

## Author Response (AR2)

**Response to reviewer #2**

We thank the reviewer for their efforts in reviewing our manuscript. Reviewer comments are shown below in black with the author response in red.

**Suggestions for revision or reasons for rejection**

The authors have primarily modified the descriptions in the manuscript, but unfortunately, these modifications are not substantial. This manuscript cites Benkiran et al. (2024; https://egusphere.copernicus.org/preprints/2024/egusphere-2024-420/), who conducted almost the same experiments and obtained similar results when comparing the impacts of assimilating SSH observations from 12 nadir satellites versus 2 swath satellites. However, their study was rejected due to a lack of novelty and unreasonable results. The same issues apply to this manuscript, as detailed below.

Benkiran et al. (2022; https://os.copernicus.org/articles/18/609/2022/) conducted experiments investigating the impacts of 2 swath satellites. This manuscript and Benkiran et al. (2024) are almost identical to Benkiran et al. (2022) except for the addition of an experiment assimilating 12 nadir satellites.

This paper differs from Benkiran et al. (2022) due to the inclusion of a comparison with the 12-nadir constellation, and the different data assimilation system used. It addresses a different question to Benkiran (2022) since it is asking the question which of two specific future constellations, proposed by a space agency, would give most impact.

The Benkiran et al. (2024) paper did indeed perform **similar** experiments to those presented here, but in fact found opposite results, i.e., the 12 nadir constellation was superior in the Met Office system, while the 2 WiSA constellation was superior in the Mercator system. As described in the introduction, these experiments were performed with different operational systems with (amongst other differences) very different data assimilation schemes. As we also described in the introduction, results from OSSEs can be very system-dependent and the aim of coordinating similar experiments with different operational systems was to allow comparison and a better understanding of the results.

The need for results from multiple systems is acknowledged by the community (see Oke & O'Kane 2011 and Fujii et al. 2019) as important in order to provide a more robust assessment of impact for future observing system design. The introduction has been updated to emphasise and clarify these points.

As mentioned in previous comments, OSSEs are useful for evaluating various observation networks that have not yet been constructed. To establish the novelty of this paper, a wide variety of experiments are needed to comprehensively investigate how the number and types of satellites contribute to analysis accuracy. However, the authors did not incorporate the comments from the first review round.

We addressed a similar comment from the reviewer in the initial review, but perhaps did not make clear enough the large computational resource required to perform these experiments. Furthermore, we addressed each of the comments raised in the initial review and updated the manuscript to include further detail of the experimental design, and further analysis of the results.

We agree that OSSEs are useful to evaluate proposed observing systems. However, due to the computational expense it was not possible for us to perform a "wide variety of experiments". Our aim here was to investigate the potential impact of two specific proposed observing networks to inform the planning of ESA as they explored options for the Sentinel-3 Next-Generation Topography mission.

The novelty here was to determine how effectively our system would be able to assimilate observations from these specific proposed networks. This allowed us to identify issues that will affect the assimilation of real wide-swath altimeter observations and so prepare to make best use of real observations as they become available.

The introduction has been updated to further clarify our aim of addressing s specific network design question and to emphasise the limitations on the number of experiments that could be run due to their computational expense.

The OSSEs showed that SSH RMSEs are 0.05-0.07 m, which is about 5-10 times larger than the prescribed SSH observation errors of 0.014 and 0.005 m for nadir and swath satellites, respectively. Since observations from 12 nadir satellites cover almost the entire global ocean, at least the RMSEs should be smaller than the observation errors. Therefore, these results are inconsistent with established data assimilation theory.

We have updated the text to clarify that the errors added to the simulated altimeter observations were intended to replicate the instrumental errors. Additionally, representation errors are introduced by the differences between the nature run and OSSE model fields. As described in section 2.3, we used the same prescribed observation errors as in our operational system which include representation errors. These vary spatially and temporally with values of 3-7cm globally.

In Section 2.4 we also describe a comparison of RMSE values for each observation type between the OSSE control and our operational system to illustrate that the results we see are similar to what we might expect to see with real observations in our operational system.

Additionally, there are still many inappropriate descriptions and terminologies, as well as insufficient details in the experimental settings and results, because the authors did not address most of the previous comments.

We addressed each of the comments raised in the initial review and updated the manuscript to include further detail of the experimental design, and further analysis of the results. Without further detail, we cannot comment here on what the reviewer still finds to be inappropriate.